# Uncertainty alters the balance between incremental learning and episodic memory

**Jonathan Nicholas[1,2]\*, Nathaniel D Daw[3,4], Daphna Shohamy[1,2,5]**

[1]Department of Psychology, Columbia University, New York, United States; [2]Mortimer B. Zuckerman Mind, Brain, Behavior Institute, Columbia University, New York, United States; [3]Department of Psychology, Princeton University, Princeton, United States; [4]Princeton Neuroscience Institute, Princeton University, Princeton, United States; [5]The Kavli Institute for Brain Science, Columbia University, New York, United States

**Abstract** A key question in decision-making is how humans arbitrate between competing learning and memory systems to maximize reward. We address this question by probing the balance between the effects, on choice, of incremental trial-and-error learning versus episodic memories of individual events. Although a rich literature has studied incremental learning in isolation, the role of episodic memory in decision-making has only recently drawn focus, and little research disentangles their separate contributions. We hypothesized that the brain arbitrates rationally between these two systems, relying on each in circumstances to which it is most suited, as indicated by uncertainty. We tested this hypothesis by directly contrasting contributions of episodic and incremental influence to decisions, while manipulating the relative uncertainty of incremental learning using a well-established manipulation of reward volatility. Across two large, independent samples of young adults, participants traded these influences off rationally, depending more on episodic information when incremental summaries were more uncertain. These results support the proposal that the brain optimizes the balance between different forms of learning and memory according to their relative uncertainties and elucidate the circumstances under which episodic memory informs decisions.

**\*For correspondence:**
jonathan.nicholas@columbia.edu

**Competing interest:** The authors declare that no competing interests exist.

## Editor's evaluation

This paper posits that higher uncertainty environments should lead to more reliance on episodic memory, finding compelling evidence for this idea across several analysis approaches and across two independent samples. This is an important paper that will be of interest to a broad group of learning, memory, and decision-making researchers.

## Introduction

Effective decision-making depends on using memories of past experiences to inform choices in the present. This process has been extensively studied using models of learning from trial-and-error, many of which rely on error-driven learning rules that in effect summarize experiences using a running average (*Sutton and Barto, 1998*; *Rescorla and Wagner, 1972*; *Houk et al., 1995*). This sort of *incremental learning* provides a simple mechanism for evaluating actions without maintaining memory traces of each individual experience along the way and has rich links to conditioning behavior and putative neural mechanisms for error-driven learning (*Schultz et al., 1997*). However, recent findings indicate that decisions may also be guided by the retrieval of individual events, a process often assumed to be supported by *episodic memory* (*Bakkour et al., 2019*; *Plonsky et al., 2015*; *Mason*

*et al., 2020*; *Bornstein et al., 2017*; *Collins and Frank, 2012*; *Bornstein and Norman, 2017*; *Duncan et al., 2019*; *Duncan and Shohamy, 2016*; *Lee et al., 2015*; *Wimmer and Büchel, 2020*). Although theoretical work has suggested a role for episodic memory in initial task acquisition, when experience is sparse (*Gershman and Daw, 2017*; *Lengyel and Dayan, 2007*), the use of episodes may be much more pervasive as its influence has been detected empirically even in decision tasks that are well-trained and can be solved normatively using incremental learning alone (*Plonsky et al., 2015*; *Bornstein et al., 2017*; *Bornstein and Norman, 2017*). The apparent ubiquity of episodic memory as a substrate for decision-making raises questions about the circumstances under which it is recruited and the implications for behavior.

How and when episodic memory is used for decisions relates to a more general challenge in cognitive control: understanding how the brain balances competing systems for decision-making. An overarching hypothesis is that the brain judiciously adopts different decision strategies in circumstances for which they are most suited; for example, by determining which system is likely to produce the most rewarding choices at the least cost. This general idea has been invoked to explain how the brain arbitrates between deliberative versus habitual decisions and previous work has suggested a key role for uncertainty in achieving a balance that maximizes reward (*Daw et al., 2005*; *Lee et al., 2014*). Moreover, imbalances in arbitration have been implicated in dysfunction such as compulsion (*Gillan et al., 2011*; *Voon et al., 2015*), addiction (*Ersche et al., 2016*; *Everitt and Robbins, 2005*), and rumination (*Hunter et al., 2022*; *Dayan and Huys, 2008*; *Huys et al., 2012*).

Here, we hypothesized that uncertainty is used for effective arbitration between decision systems and tested this hypothesis by investigating the tradeoff between incremental learning and episodic memory. This is a particularly favorable setting in which to examine this hypothesis due to a rich prior literature theoretically analyzing, and experimentally manipulating, the efficacy of incremental learning in isolation. Studies of this sort typically manipulate the volatility, or frequency of change, of the environment, as a way of affecting uncertainty about incrementally learned quantities. In line with predictions made by statistical learning models, these experiments demonstrate that when the reward associated with an action is more volatile, people adapt by increasing their incremental learning rates (*Behrens et al., 2007*; *Mathys et al., 2011*; *O'Reilly, 2013*; *Nassar et al., 2012*; *Nassar et al., 2010*; *Browning et al., 2015*; *Piray and Daw, 2020*; *Kakade and Dayan, 2002*; *Yu and Dayan, 2005*). In this case, incrementally constructed estimates reflect a running average over fewer experiences, yielding both less accurate and more uncertain estimates of expected reward. We, therefore, reasoned that the benefits of incremental learning are most pronounced when incremental estimation can leverage many experiences or, in other words, when volatility is low. By contrast, when the environment is either changing frequently or has recently changed, estimating reward episodically by retrieving a single, well-matched experience should be relatively more favorable.

We tested this hypothesis using a choice task that directly pits these decision systems against one another (*Duncan et al., 2019*), while manipulating volatility. In particular, we (i) independently measured the contributions of episodic memory vs. incremental learning to choice and (ii) altered the uncertainty about incremental estimates using different levels of volatility. Two large online samples of healthy young adults completed three tasks. Results from the primary sample (n = 254) are reported in the main text; results from a replication sample (n = 223) are reported in the appendices (Appendix 1).

The main task of interest combined incremental learning and episodic memory, referred to throughout as the *deck learning and card memory* task (*Figure 1A*, middle panel). On each trial of this task, participants chose between two cards of a different color and received feedback following their choice. The cards appeared on each trial throughout the task, but their relative value changed over time (*Figure 1B*). In addition to the color of the card, each card also displayed an object. Critically, objects appeared on a card at most twice throughout the task, such that a chosen object could reappear between 9 and 30 trials after it was chosen the first time, and would deliver the same reward. Thus, participants could make decisions based on incremental learning of the average value of the decks or based on episodic memory for the specific value of an object which they only saw once before. Additionally, participants made choices across two environments: a *high-volatility* and a *low-volatility* environment. The environments differed in how often reversals in deck value occurred.

In addition to the main task, participants also completed two other tasks in the experiment. First, participants completed a simple *deck learning* task (*Figure 1A*, left panel) to acclimate them to each environment and quantify the effects of uncertainty. This task included choices between two diamonds

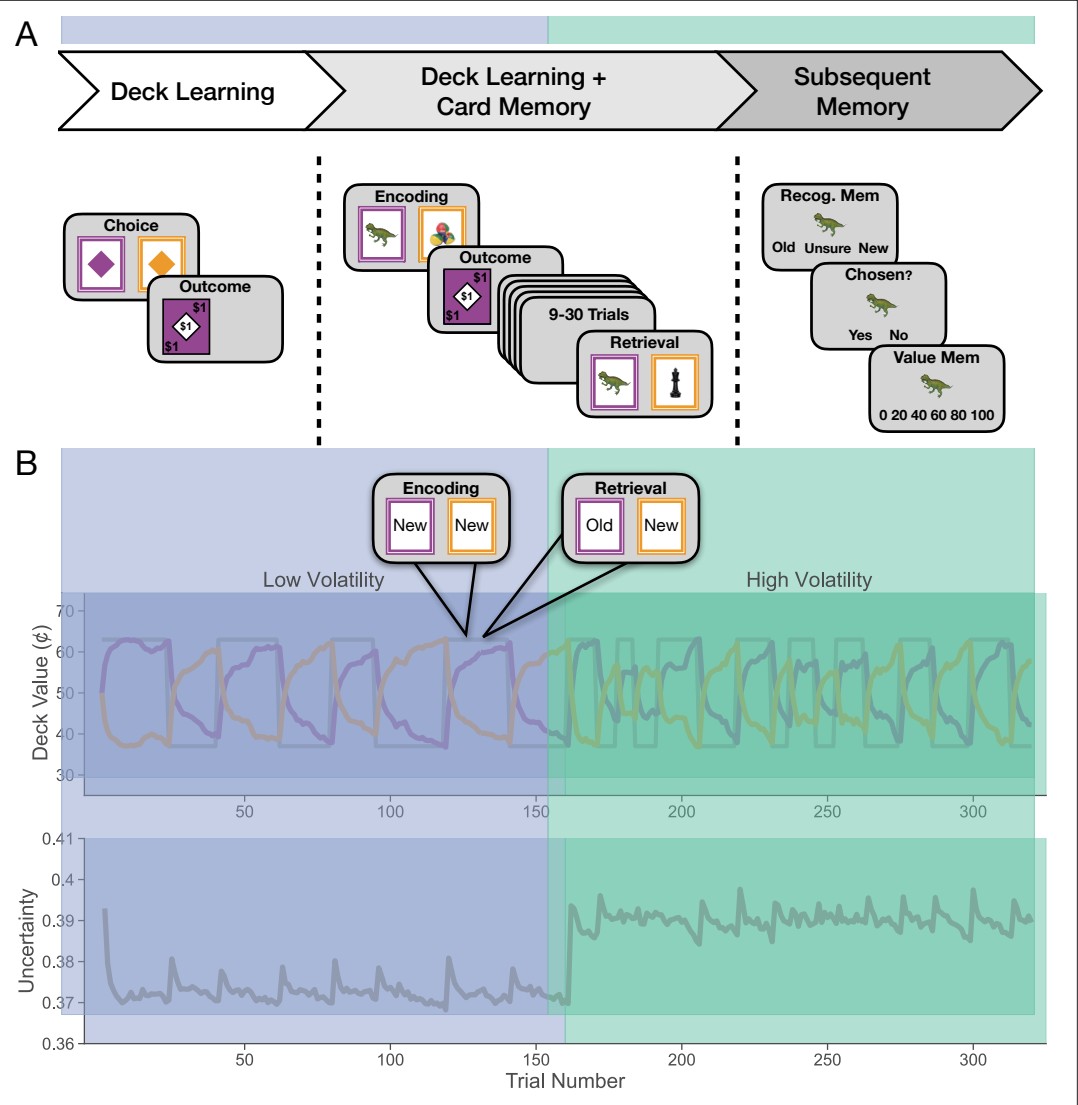

**Figure 1.** Study design and sample events. (**A**) Participants completed three tasks in succession. The first was the *deck learning* task that consisted of choosing between two colored cards and receiving an outcome following each choice. One color was worth more on average at any given timepoint, and this mapping changed periodically. Second was the main task of interest, the *deck learning and card memory* task, which followed the same structure as the deck learning task but each card also displayed a trial-unique object. Cards that were chosen could appear a second time in the task after 9–30 trials and, if they reappeared, were worth the same amount, thereby allowing participants to use episodic memory for individual cards in addition to learning deck value from feedback. Outcomes ranged from $0 to $1 in increments of 20¢ in both of these tasks. Lastly, participants completed a *subsequent memory* task for objects that may have been seen in the deck learning and card memory task. Participants had to indicate whether they recognized an object and, if they did, whether they chose that object. If they responded that they had chosen the object, they were then asked if they remembered the value of that object. (**B**) Uncertainty manipulation within and across environments. Uncertainty was manipulated by varying the volatility of the relationship between cue and reward over time. Participants completed the task in two counterbalanced environments that differed in their relative volatility. The low-volatility environment featured half as many reversals in deck luckiness as the high-volatility environment. *Top:* the true value of the purple deck is drawn in gray for an example trial sequence. In purple and orange are estimated deck values from the reduced Bayesian model (*Nassar et al., 2010*). Trials featuring objects appeared only in the deck learning and card memory task. *Bottom:* uncertainty about deck value as estimated by the model is shown in gray. This plot shows relative uncertainty, which is the model's imprecision in its estimate of deck value.

of a different color on each trial, without any trial-unique objects. Second, after the main task, participants completed a standard *subsequent memory* task (*Figure 1A*, right panel) designed to assess later episodic memory for objects encountered in the main task.

We predicted that greater uncertainty about incremental values would be related to increased use of episodic memory. The experimental design provided two opportunities to measure the impact of uncertainty: *across* conditions, by comparing between the high- and the low-volatility environments, and *within* condition, by examining how learning and choices were impacted by each reversal.

## Results

### Episodic memory is used more under conditions of greater volatility

As noted above, participants completed two decision-making tasks. The *deck learning* task familiarized them with the underlying incremental learning task and established an independent measure of sensitivity to the volatility manipulation. The separate *deck learning and card memory* task measured the additional influence of episodic memory on decisions (*Figure 1*). In the deck learning task, participants chose between two decks with expected value ($V$) that reversed periodically across two environments, with one more volatile (reversals every 10 trials on average) and the other less volatile (reversals every 20 trials on average).

Participants were told that at any point in the experiment one of the two decks was 'lucky,' meaning that its expected value ($V_{lucky}$ = 63¢) was higher than the other 'unlucky' deck ($V_{unlucky}$ = 37¢). They were also told that which deck was currently lucky could reverse at any time, and that they would be completing the task in two environments that differed in how often these reversals occurred. We reasoned that, following each reversal, participants should be more uncertain about deck value and

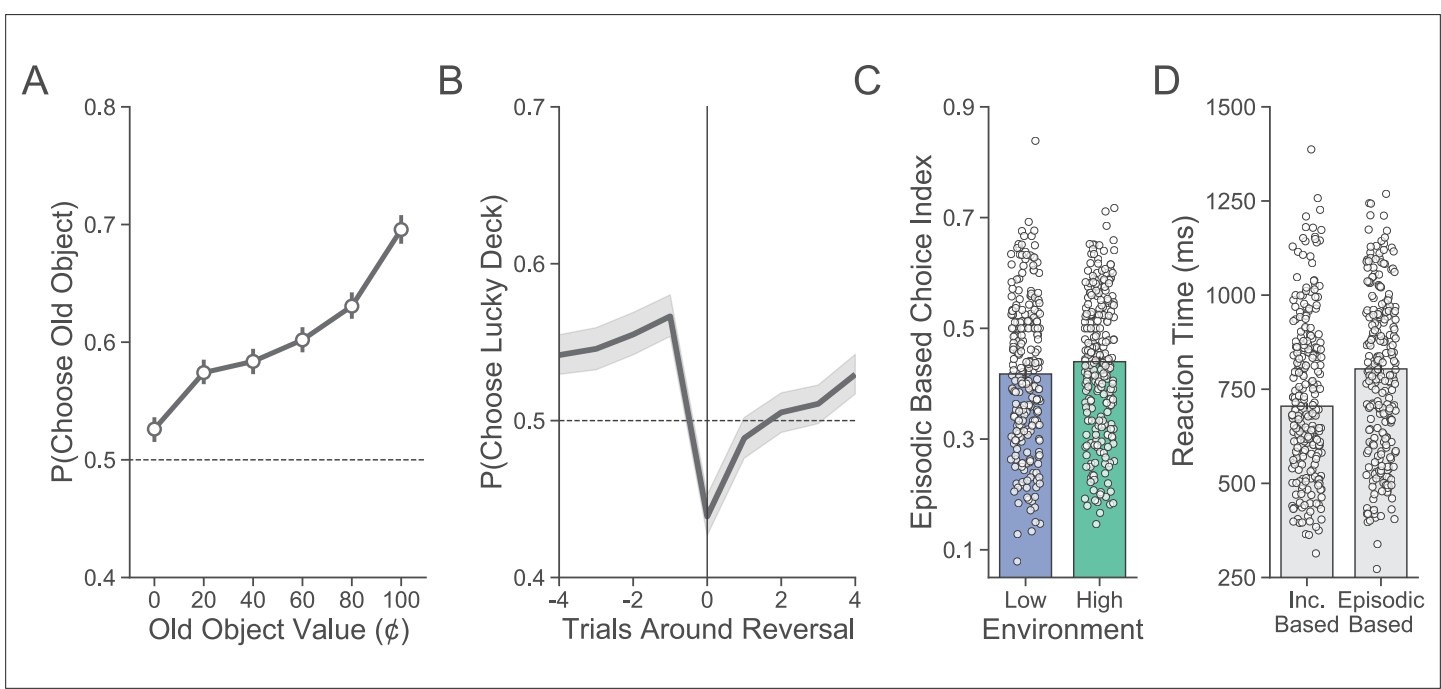

**Figure 2.** Evaluating the proportion of incremental and episodic choices. (**A**) Participants' (n = 254) choices demonstrate sensitivity to the value of old objects. Group-level averages are shown as points and lines represent 95% confidence intervals. (**B**) Reversals in deck luckiness altered choice such that the currently lucky deck was chosen less following a reversal. The line represents the group-level average, and the band represents the 95% confidence interval. (**C**) On incongruent trials, choices were more likely to be based on episodic memory (e.g., high-valued objects chosen and low-valued objects avoided) in the high- compared to the low-volatility environment. Averages for individual subjects are shown as points, and lines represent the group-level average with a 95% confidence interval. (**D**) Median reaction time was longer for incongruent choices based on episodic memory compared to those based on incremental learning.

The online version of this article includes the following figure supplement(s) for figure 2:

**Figure supplement 1.** Recreation of *Figure 2* in the main text using the replication dataset.

that this uncertainty should reduce with experience. Because the more volatile environment featured more reversals, in this condition subjects should have greater uncertainty about the deck value overall.

In the second deck learning and card memory task, each deck featured cards with trial-unique objects that could reappear once after being chosen and were worth an identical amount at each appearance. Here, participants were told that they could use their memory for the value of objects they recognized to guide their choices. They were also told that the relative level of volatility in each environment during the card learning task would be identical in this task. We predicted that decisions would be based more on object value when deck value was more volatile. Our logic was that episodic memory should be relied upon more strongly when incremental learning is less accurate and reliable due to frequent change. This, in turn, is because episodic memory is itself imperfect in practice, so participants face a nontrivial tradeoff between attempting episodic recall vs. relying on incremental learning when an object recurs. We, therefore, expected choices to be more reliant on episodic memory in the high- compared to the low-volatility environment.

We first examined whether participants were separately sensitive to each source of value in the deck learning and card memory task: the value of the objects (episodic) and of the decks (incremental). Controlling for average deck value, we found that participants used episodic memory for object value, evidenced by a greater tendency to choose high-valued old objects than low-valued old objects ($\beta_{OldValue} = 0.621, 95\%CI = [0.527, 0.713]$; *Figure 2A*). Likewise, controlling for object value, we also found that participants used incrementally learned value for the decks, evidenced by the fact that the higher-valued (lucky) deck was chosen more frequently on trials immediately preceding a reversal ($\beta_{t-4} = 0.038, 95\%\ CI = [-0.038, 0.113]$; $\beta_{t-3} = 0.056, 95\%\ CI = [-0.02, 0.134]$; $\beta_{t-2} = 0.088, 95\%\ CI = [0.009, 0.166]$; $\beta_{t-1} = 0.136, 95\%\ CI = [0.052, 0.219]$; *Figure 2B*), that this tendency was disrupted by the reversals ($\beta_{t=0} = -0.382, 95\%CI = [-0.465, -0.296]$), and by the quick recovery of performance on the trials following a reversal ($\beta_{t+1} = -0.175, 95\%CI = [-0.258, -0.095]$; $\beta_{t+2} = -0.106, 95\%CI = [-0.18, -0.029]$; $\beta_{t+3} = -0.084, 95\%CI = [-0.158, -0.006]$; $\beta_{t+4} = 0.129, 95\%CI = [0.071, 0.184]$).

Having established that both episodic memory and incremental learning guided choices, we next sought to determine the impact of volatility on episodic memory for object value by isolating trials on which episodic memory was most likely to be used. To identify reliance on object value, we first focused on trials where the two sources of value information were incongruent: that is, trials for which the high-value deck featured an old object that was of low value (<50¢) or the low-value deck featured an old object that was of high value (>50¢). We then defined an *episodic-based choice index* (EBCI) by considering a choice as episodic if the old object was, in the first case, avoided or, in the second case, chosen. Consistent with our hypothesis, we found greater evidence for episodic choices in the high-volatility environment compared to the low-volatility environment ($\beta_{Env} = 0.092, 95\%CI = [0.018, 0.164]$; *Figure 2C*). Finally, this analysis also gave us the opportunity to test differences in reaction time between incremental and episodic decisions. Decisions based on episodic value took longer ($\beta_{EBCI} = 37.629, 95\%CI = [28.488, 46.585]$; *Figure 2D*), perhaps reflecting that episodic retrieval may take more time than retrieval of cached incremental value.

## Uncertainty about incremental values increases sensitivity to episodic value

The effects of environment described above provide a coarse index of overall differences in learning across conditions. To capture uncertainty about deck value on a trial-by-trial basis, we adopted a computational model that tracks uncertainty during learning. We then used this model to test our central hypothesis: that episodic memory is used more when posterior uncertainty about deck value is high. Our reasoning was that episodic memory should not only be deployed more when incremental learning is overall inaccurate due to frequent change, but also within either condition following recent change. We, therefore, predicted that, across both environments, participants would be more likely to recruit episodic memory following reversals in deck value, when uncertainty is at its highest.

We began by hierarchically fitting two classes of incremental learning models to the behavior on the deck learning task: a baseline model with a Rescorla–Wagner (*Rescorla and Wagner, 1972*) style update (RW) and a reduced Bayesian model (*Nassar et al., 2010*) (RB) that augments the RW learner with a variable learning rate, which it modulates by tracking ongoing uncertainty about deck value. This approach – which builds on a line of work applying Bayesian learning models to capture trial-by-trial modulation in uncertainty and learning rates in volatile environments (*Behrens et al., 2007*;

*Mathys et al., 2011*; *Nassar et al., 2010*; *Piray and Daw, 2020*; *Kakade and Dayan, 2002*; *Yu and Dayan, 2005*) – allowed us to first assess incremental learning free of any contamination due to competition with episodic memory. We then used the parameters fit to this task for each participant to generate estimates of subjective deck value and uncertainty around deck value, out of sample, in the deck learning and card memory task. These estimates were then used alongside episodic value to predict choices on incongruent trials in the deck learning and card memory task.

We first tested whether participants adjusted their rates of learning in response to uncertainty, both between environments and due to trial-wise fluctuations in uncertainty about deck value. We did this by comparing the ability of each combined choice model to predict participants' decisions out of sample. To test for effects between environments, we compared models that controlled learning with either a single free parameter (for RW, a learning rate $\alpha$; for RB, a hazard rate $H$ capturing the expected frequency of reversals) shared across both environments or models with a separate free parameter for each environment. To test for trial-wise effects within environments, we compared between RB and RW models: while RW updates deck value with a constant learning rate, RB tracks ongoing posterior uncertainty about deck value (called relative uncertainty, RU) and increases its learning rate when this quantity is high.

We also included two other models in our comparison to control for alternative learning strategies. The first was a contextual inference model (CI), which modeled deck value as arising from two switching contexts (either that one deck was lucky and the other unlucky or vice versa) rather than from incremental learning. The second was a Rescorla–Wagner model that, like the RB model but unlike the RW models described above, learned only a single-value estimate (RW1Q). The details for all models can be found in Appendix 3.

Participants were sensitive to the volatility manipulation and also incorporated uncertainty into updating their beliefs about deck value. This is indicated by the fact that the RB combined choice model that included a separate hazard rate for each environment (RB2$H$) outperformed both RW models, the RB model with a single hazard rate, as well as other alternative learning models (*Figure 3A*). Further, across the entire sample, participants detected higher levels of volatility in the high-volatility environment, as indicated by the generally larger hazard rates recovered from this model in the high- compared to the low-volatility environment ($H_{Low} = 0.04, 95\%CI = [0.033, 0.048]$; $H_{High} = 0.081, 95\%CI = [0.067, 0.097]$; *Figure 3B*). Next, we examined the model's ability to estimate uncertainty as a function of reversals in deck luckiness. Compared to an average of the four trials prior to a reversal, RU increased immediately following a reversal and stabilized over time ($\beta_{t=0} = 0.014, 95\%CI = [-0.019, 0.048]$; $\beta_{t+1} = -0.242, 95\%CI = [-0.276, -0.209]$; $\beta_{t+2} = -0.145, 95\%CI = [-0.178, -0.112]$; $\beta_{t+3} = -0.1, 95\%CI = [-0.131, -0.07]$; $\beta_{t+4} = -0.079, \ 95\%CI = [-0.108, -0.048]$; *Figure 3C*). As expected, RU was also, on average, greater in the high- compared to the low-volatility environment ($\beta_{Env} = 0.015$, $95\%CI = [0.012, 0.018]$). Lastly, we were interested in assessing the relationship between reaction time and RU as we expected that higher uncertainty may be reflected in more time needed to resolve decisions. In line with this idea, RU was strongly related to reaction time such that choices made under more uncertain conditions took longer ($\beta_{RU} = 1.685, 95\%CI = [0.823, 2.528]$).

Having established that participants were affected by uncertainty around beliefs about deck value, we turned to examine our primary question: whether this uncertainty alters the use of episodic memory in choices. We first examined effects of RU on the episodic choice index, which measures choices consistent with episodic value on trials when it disagrees with incremental learning. This analysis verified that episodic memory was used more on incongruent trial decisions made under conditions of high RU ($\beta_{RU} = 2.133, 95\%CI = [0.7, 3.535]$; *Figure 4A*). To more directly test the prediction that participants would use episodic memory when uncertainty is high, we included trial-by-trial estimates of RU in the RB2$H$ combined choice model, which was augmented with an additional free parameter to capture any change with RU in the effect of episodic value on choice. Formally, this parameter measured an effect of the interaction between these two factors, and the more positive this term the greater the impact of increased uncertainty on the use of episodic memory. This new combined choice model further improved out-of-sample predictions (RB2$H$+RU, *Figure 3A*). As predicted, while both incremental and episodic value were used overall ($\beta_{DeckValue} = 0.502, 95\%CI = [0.428, 0.583]$; $\beta_{OldValue} = 0.150, 95\%CI = [0.101, \ 0.20]$), episodic value impacted choices more when RU was high ($\beta_{OldValue:RU} = 0.067, 95\%CI = [0.026, 0.11]$; *Figure 4B*) and more generally in the high- compared

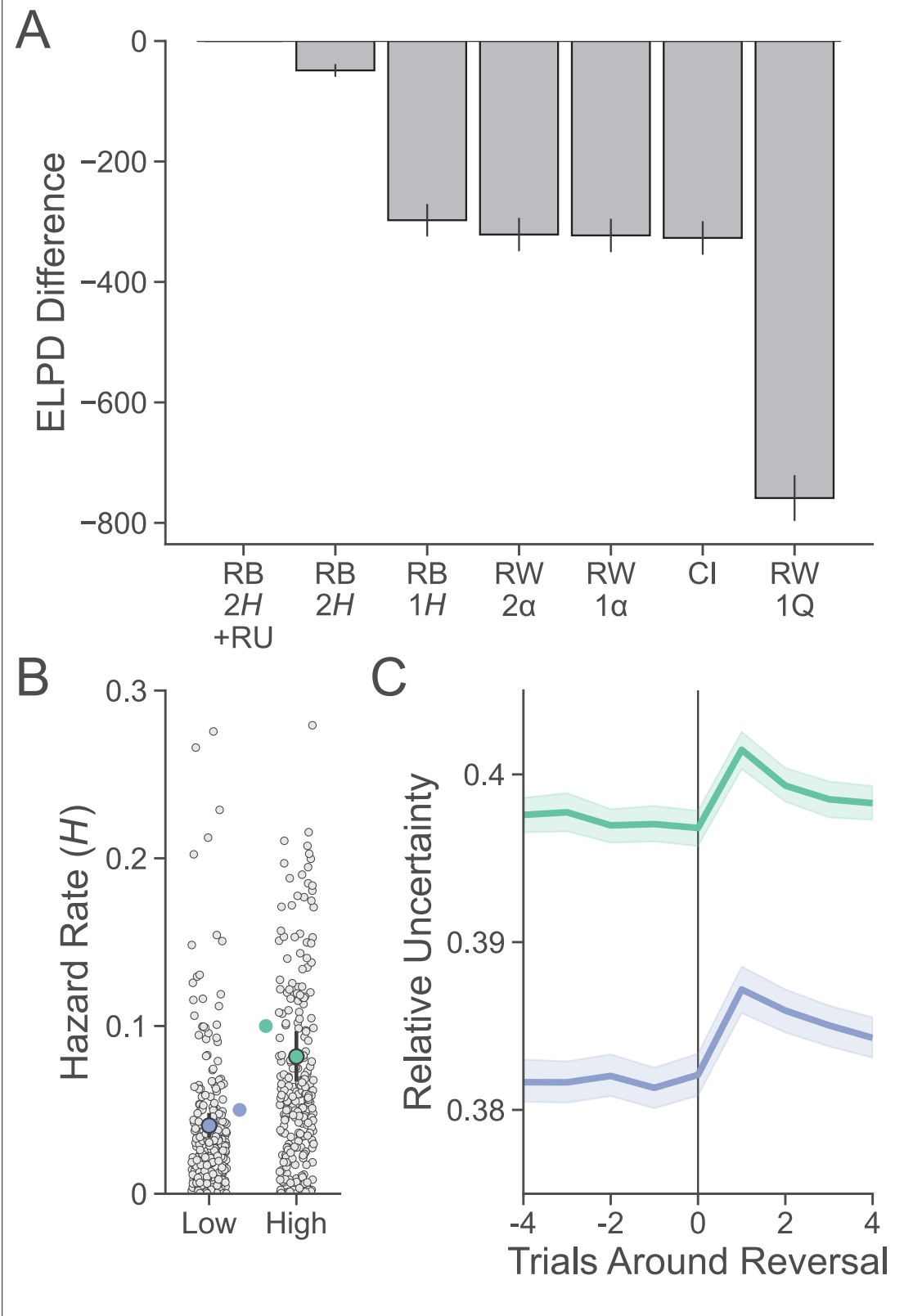

**Figure 3.** Evaluating model fit and sensitivity to volatility. (**A**) Expected log pointwise predictive density (ELPD) from each model was calculated from a 20-fold leave-N-subjects-out cross-validation procedure and is shown here subtracted from the best-fitting model. The best-fitting model was the reduced Bayesian (RB) model with two hazard rates (2H) and sensitivity to the interaction between old object value and relative uncertainty (RU) in the choice function. Error bars represent standard error around ELPD estimates. (**B**) Participants (n = 254) were sensitive to the relative level of volatility in

*Figure 3 continued on next page*

*Figure 3 continued*

each environment as measured by the hazard rate. Group-level parameters are superimposed on individual subject parameters. Error bars represent 95% posterior intervals. The true hazard rate for each environment is shown on the interior of the plot. (**C**) RU peaks on the trial following a reversal and is greater in the high- compared to the low-volatility environment. Lines represent group means, and bands represent 95% confidence intervals.

The online version of this article includes the following figure supplement(s) for figure 3:

**Figure supplement 1.** Recreation of *Figure 3* in the main text using the replication dataset.

to the low-volatility environment ($\beta_{OldValue:\,Env} = 0.06, 95\%CI = [0.02, 0.1]$). This is consistent with the hypothesis that episodic value was relied on more when beliefs about incremental value were uncertain.

The analyses above focus on uncertainty present at the time of retrieving episodic value because this is what we hypothesized would drive competition in the reliance on either system at choice time. However, in principle, reward uncertainty at the time an object is first encountered might also affect its encoding, and hence its subsequent use in episodic choice when later retrieved (*Rouhani et al., 2018*). To address this possibility, we looked at the impact of RU resulting from the first time an old object's value was revealed on whether that object was later retrieved for a decision. Using our EBCI, there was no relationship between the use of episodic memory on incongruent trial decisions and RU at encoding ($\beta_{RU} = 0.622, 95\%CI = [-0.832, 2.044]$; *Figure 4—figure supplement 2A*). Similarly, we also examined effects of trial-by-trial estimates of RU at encoding time in the combined choice model by adding another free parameter that captured change with RU at encoding time in the effect of episodic value on choice. This parameter was added alongside the effect of RU at retrieval time (from the previous analysis). There was no effect on choice in either sample (main: $\beta_{OldValue:\,RU} = 0.028, 95\%CI = [-0.011, 0.067]$; replication: $\beta_{OldValue:\,RU} = -0.003, 95\%CI = [-0.046, 0.037]$; *Figure 4—figure supplement 2B*) and the

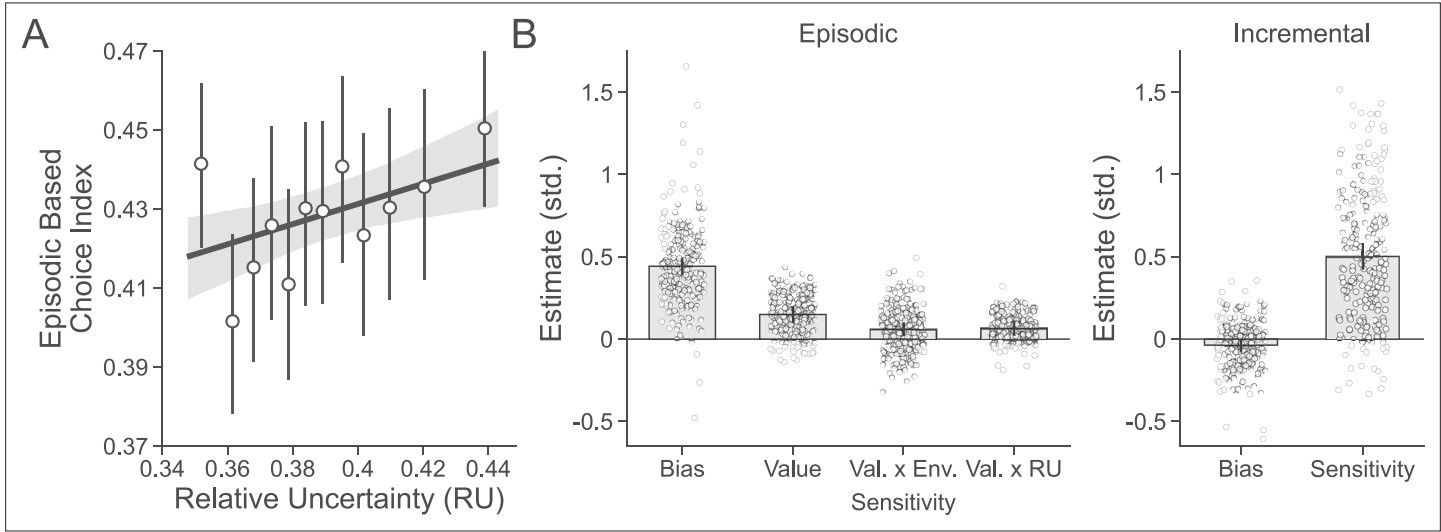

**Figure 4.** Evaluating effects of sensitivity to uncertainty on episodic choices. (**A**) Participants' (n = 254) degree of episodic-based choice increased with greater relative uncertainty (RU) as predicted by the combined choice model. Points are group means, and error bars are 95% confidence intervals. (**B**) Estimates from the combined choice model. Participants were biased to choose previously seen objects regardless of their value and were additionally sensitive to their value. As hypothesized, this sensitivity was increased when RU was higher, as well as in the high- compared to the low-volatility environment. There was no bias to choose one deck color over the other, and participants were highly sensitive to estimated deck value. Group-level parameters are superimposed as bars on individual subject parameters represented as points. Error bars represent 95% posterior intervals around group-level parameters. Estimates are shown in standard units.

The online version of this article includes the following figure supplement(s) for figure 4:

**Figure supplement 1.** Recreation of *Figure 4* in the main text using the replication dataset.

**Figure supplement 2.** Results of relative uncertainty (RU) at encoding time on episodic-based choice in the main (**A–C**) and replication (**D–F**) sample.

inclusion of this parameter did not provide a better fit to subjects' choices than the combined choice model with only increased sensitivity due to RU at retrieval time (*Figure 4—figure supplement 2C*).

## Episodic and incremental value sensitivity predicts subsequent memory performance

Having determined that decisions depended on episodic memory more when uncertainty about incremental value was higher, we next sought evidence for similar effects on the quality of episodic memory. Episodic memory is, of course, imperfect, and value estimates derived from episodic memory are therefore also uncertain. More uncertain episodic memory should then be disfavored while the influence of incremental value on choice is promoted instead. Although in this study we did not experimentally manipulate the strength of episodic memory, as our volatility manipulation was designed to affect the uncertainty of incremental estimates, we did measure memory strength in a subsequent memory test. Thus, we predicted that participants who base fewer decisions on object value and more decisions on deck value should have poorer subsequent memory for objects from the deck learning and card memory task.

We first assessed subsequent memory performance. Participants' recognition memory was well above chance ($\beta_0 = 1.887, 95\%CI = [1.782, 1.989]$), indicating a general ability to discriminate objects seen in the main task from those that were new. Recall for the value of previously seen objects was also well predicted by their true value ($\beta_{TrueValue} = 0.174, 95\%CI = [0.160, 0.188]$), providing further support that episodic memory was used to encode object value. To underscore this point, we sorted subsequent memory trials according to whether an object was seen on an episodic- or incremental-based choice, as estimated according to our EBCI, during the deck learning and card memory task. Not only were objects from episodic-based choices better remembered than those from incremental-based choices ($\beta_{EBCI} = 0.192, 95\%CI = [0.072, 0.322]$; *Figure 5A*), but value recall was also improved for these objects ($\beta_{EBCI:\ TrueValue} = 0.047, 95\%CI = [0.030, 0.065]$; *Figure 5B*).

We next leveraged the finer-grained estimates of sensitivity to episodic value from the learning model to ask whether, across participants, individuals who were estimated to deploy episodic value more during the deck learning and card memory task also performed better on the subsequent memory test. In line with the idea that episodic memory quality also impacts the relationship between incremental learning and episodic memory, participants with better subsequent recognition memory were more sensitive to episodic value ($\beta_{EpSensitivity} = 0.373, 95\%CI = [0.273, 0.478]$; *Figure 5C*), and these same participants were less sensitive to incremental value ($\beta_{IncSensitivity} = -0.276, 95\%CI = [-0.383, -0.17]$; *Figure 5D*). This result provides further evidence for a tradeoff between episodic memory and incremental learning. It also provides preliminary support for a broader version of our hypothesis, which is that uncertainty about value provided by either memory system arbitrates the balance between them.

Lastly, the subsequent memory task also provided us with the opportunity to replicate other studies that have found that prediction error and its related quantities enhance episodic memory across a variety of tasks and paradigms (*Rouhani et al., 2018*; *Rouhani and Niv, 2021*; *Antony et al., 2021*; *Ben-Yakov et al., 2022*). We predicted that participants should have better subsequent memory for objects encoded under conditions of greater uncertainty. While not our primary focus, we found support for this prediction across both samples (see Appendix 2, *Figure 5—figure supplement 2*).

## Replication of the main results in a separate sample

We repeated the tasks described above in an independent online sample of healthy young adults (n = 223) to test the replicability and robustness of our findings. We replicated all effects of environment and RU on episodic-based choice and subsequent memory (see Appendix 1 and figure supplements for details).

## Discussion

Research on learning and value-based decision-making has focused on how the brain summarizes experiences by error-driven incremental learning rules that, in effect, maintain the running average of many experiences. While recent work has demonstrated that episodic memory also contributes to value-based decisions (*Bakkour et al., 2019*; *Plonsky et al., 2015*; *Mason et al., 2020*; *Bornstein et al., 2017*; *Collins and Frank, 2012*; *Bornstein and Norman, 2017*; *Duncan et al., 2019*; *Duncan*

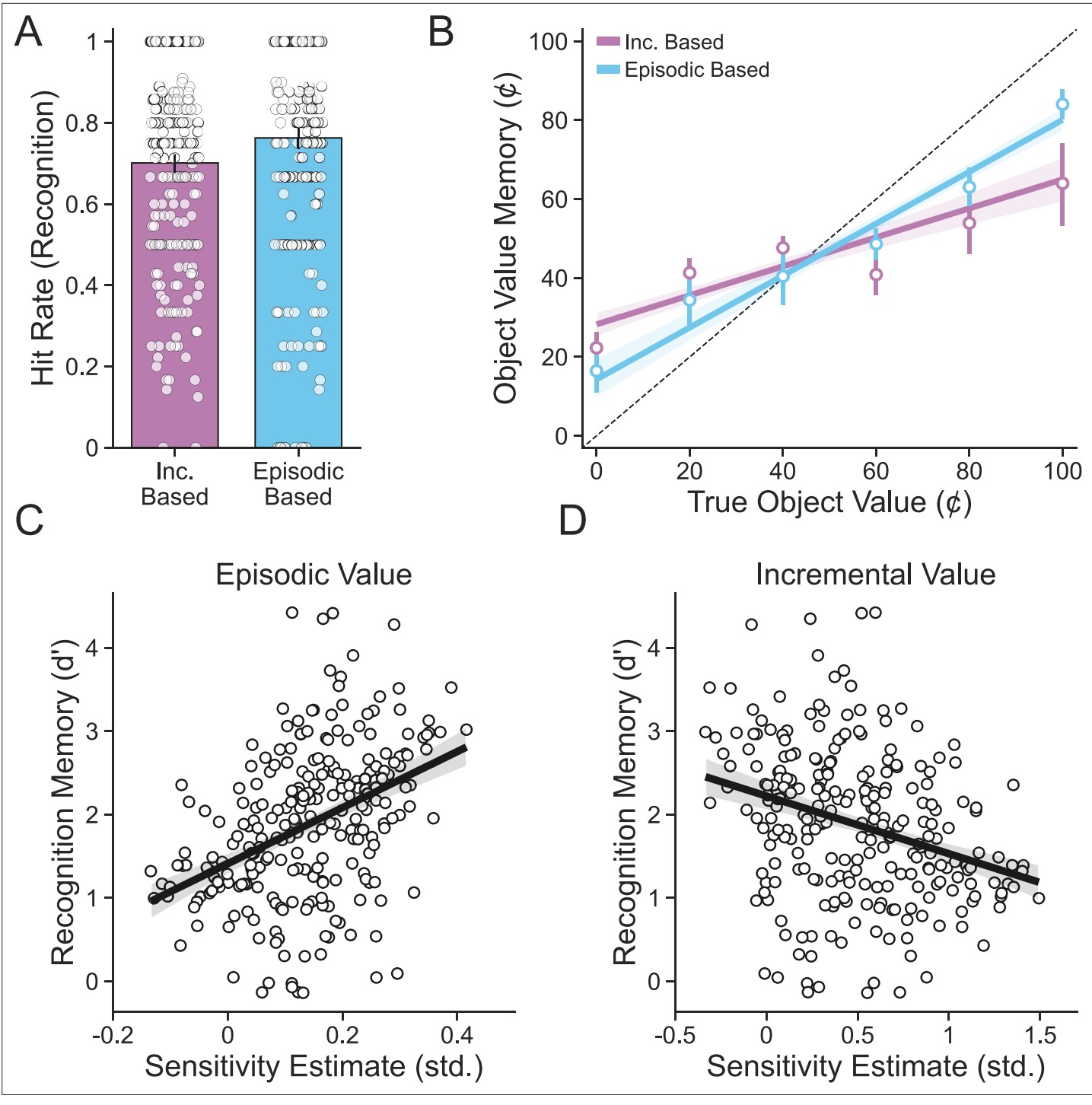

**Figure 5.** Relationship between choice type and subsequent memory. (**A**) Objects originally seen during episodic-based choices were better remembered than objects seen during incremental-based choices. Average hit rates for individual subjects (n = 254) are shown as points, bars represent the group-level average, and lines represent 95% confidence intervals. (**B**) The value of objects originally seen during episodic-based choices was better recalled than objects seen during incremental-based choices. Points represent average value memory for each possible object value, and error bars represent 95% confidence intervals. Lines are linear fits, and bands are 95% confidence intervals. (**C**) Participants with greater sensitivity to episodic value as measured by random effects in the combined choice model tended to better remember objects seen originally in the card learning and deck memory task. (**D**) Participants with greater sensitivity to incremental value tended to have worse memory for objects from the card learning and deck memory task. Points represent individual participants, lines are linear fits, and bands are 95% confidence intervals.

The online version of this article includes the following figure supplement(s) for figure 5:

*Figure 5 continued on next page*

*Figure 5 continued*

**Figure supplement 1.** Recreation of *Figure 5* in the main text using the replication dataset.

**Figure supplement 2.** Effects of relative uncertainty (RU), changepoint probability (CPP), and absolute prediction error (APE) at encoding time on subsequent recognition and value memory in both the main and replication samples.

*and Shohamy, 2016*; *Lee et al., 2015*; *Wimmer and Büchel, 2020*), many open questions remain about the circumstances under which episodic memory is used. We used a task that directly contrasts episodic and incremental influences on decisions and found that participants traded these influences off rationally, relying more on episodic information when incremental summaries were less reliable, that is, more uncertain and based on fewer experiences. We also found evidence for a complementary modulation of this episodic-incremental balance by episodic memory quality, suggesting that more uncertain episodic-derived estimates may reduce reliance on episodic value. Together, these results indicate that reward uncertainty modulates the use of episodic memory in decisions, suggesting that the brain optimizes the balance between different forms of learning according to volatility in the environment.

Our findings add empirical data to previous theoretical and computational work, which has suggested that decision-making can greatly benefit from episodic memory for individual estimates when available data are sparse. This most obviously arises early in learning a new task, but also in task transfer, high-dimensional or non-Markovian environments, and (as demonstrated in this work) during conditions of rapid change (*Lengyel and Dayan, 2007*; *Blundell, 2016*; *Santoro et al., 2016*). We investigate these theoretical predictions in the context of human decision-making, testing whether humans rely more heavily on episodic memory when incremental summaries comprising multiple experiences are relatively poor. We operationalize this tradeoff in terms of uncertainty, exemplifying a more general statistical scheme for arbitrating between different decision systems by treating them as estimators of action value.

There is precedent for this type of uncertainty-based arbitration in the brain, with the most well-known being the tradeoff between model-free learning and model-based learning (*Daw et al., 2005*; *Keramati et al., 2011*). Control over decision-making by model-free and model-based systems has been found to shift in accordance with the accuracy of their respective predictions (*Lee et al., 2014*), and humans adjust their reliance on either system in response to external conditions that provide a relative advantage to one over the other (*Simon and Daw, 2011*; *Kool et al., 2016*; *Otto et al., 2013*). Tracking uncertainty provides useful information about when inaccuracy is expected and helps to maximize utility by deploying whichever system is best at a given time. Our results add to these findings and expand their principles to include episodic memory in this tradeoff. This may be especially important given that human memory is resource limited and prone to distortion (*Schacter et al., 2011*) and forgetting (*Ebbinghaus, 2013*). Notably, in our task, an observer equipped with perfect episodic memory would always benefit from using it to make decisions. Yet, as our findings show, participants vary in their episodic memory abilities, and this memory capacity is related to the extent to which episodic memory is used to guide decisions.

One intriguing possibility is that there is more than just an analogy between the incremental-episodic balance studied here and previous work on model-free versus model-based competition. Incremental error-driven learning coincides closely with model-free learning in other settings (*Schultz et al., 1997*; *Daw et al., 2005*) and, although it has been proposed that episodic control constitutes a 'third way' (*Lengyel and Dayan, 2007*), it is possible that behavioral signatures of model-based learning might instead arise from episodic control via covert retrieval of individual episodes (*Gershman and Daw, 2017*; *Hassabis and Maguire, 2009*; *Schacter et al., 2012*; *Vikbladh et al., 2017*), which contain much of the same information as a cognitive map or world model. While this study assesses single-event episodic retrieval more overtly, an open question for future work is whether the extent to which these same processes, and ultimately the same episodic-incremental tradeoff, might also explain model-based choice as it has been operationalized in other decision tasks. A related line of work has emphasized a similar role for working memory in maintaining representations of individual trials for choice (*Collins and Frank, 2012*; *Yoo and Collins, 2022*; *Collins, 2018*; *Collins and Frank, 2018*). Given the capacity constraints of working memory, we think it unlikely that working memory can account for the effects shown here, which involve memory for dozens of trial-unique stimuli maintained over tens of trials.

Our findings also help clarify the impacts of uncertainty, novelty, and prediction error on episodic memory. Recent studies found that new episodes are more likely to be encoded under novel circumstances while prior experiences are more likely to be retrieved when conditions are familiar (*Duncan et al., 2019*; *Duncan and Shohamy, 2016*; *Duncan et al., 2012*; *Hasselmo, 2006*). Shifts between these states of memory are thought to be modulated by one's focus on internal or external sources of information (*Decker and Duncan, 2020*; *Tarder-Stoll et al., 2020*) and signaled by prediction errors based in episodic memory (*Bein et al., 2020*; *Chen et al., 2015*; *Sinclair and Barense, 2018*; *Greve et al., 2017*). Relatedly, unsigned prediction errors, which are a marker of surprise, improve later episodic memory (*Rouhani et al., 2018*; *Rouhani and Niv, 2021*; *Antony et al., 2021*; *Ben-Yakov et al., 2022*). Findings have even suggested that states of familiarity and novelty can bias decisions toward the use of single past experiences or not (*Duncan et al., 2019*; *Duncan and Shohamy, 2016*).

One alternative hypothesis that emerges from this work is that change-induced uncertainty and novelty could exert similar effects on memory, such that novelty signaled by expectancy violations increases encoding in a protracted manner that dwindles as uncertainty is resolved, or the state of the environment becomes familiar. Our results provide mixed support for this interpretation. While subsequent memory was improved by the presence of uncertainty at encoding, as would be predicted by this work, there was little effect of uncertainty at encoding time on the extent to which decisions were guided by individual memories. It, therefore, seems likely that uncertainty and novelty operate in concert but exert different effects over decision-making, an interpretation supported by recent evidence (*Xu et al., 2021*).

This work raises further questions about the neurobiological basis of memory-based decisions and the role of neuromodulation in signaling uncertainty and aiding memory. In particular, studies have revealed unique functions for norepinephrine (NE) and acetylcholine (ACh) on uncertainty and learning. These findings suggest that volatility, as defined here, is likely to impact the noradrenergic modulatory system, which has been found to signal unexpected changes throughout learning (*Nassar et al., 2012*; *Yu and Dayan, 2005*; *Yu and Dayan, 2002*; *Zhao et al., 2019*). Noradrenergic terminals densely innervate the hippocampus (*Schroeter et al., 2000*), and a role for NE in both explicit memory formation (*Grella et al., 2019*) and retrieval (*Murchison et al., 2004*) has been posited. Future studies involving a direct investigation of NE or an indirect investigation using pupillometry (*Nassar et al., 2012*) may help to isolate its contributions to the interaction between incremental learning and episodic memory in decision-making. ACh is also important for learning and memory as memory formation is facilitated by ACh in the hippocampus, which may contribute to its role in separating and storing new experiences (*Hasselmo, 2006*; *Decker and Duncan, 2020*). In addition to this role, ACh is heavily involved in incremental learning and has been widely implicated in signaling expected uncertainty (*Yu and Dayan, 2002*; *Bland and Schaefer, 2012*). ACh may therefore play an important part in managing the tradeoff between incremental learning and episodic memory.

Indeed, while in this work we investigated the impact of uncertainty on learning using a well-established manipulation of environmental volatility, in general (and even in this task) uncertainty also arises from many other parameters of the environment, such as stochasticity (trial-wise outcome variance) (*Piray and Daw, 2021*). It remains to be seen whether similar results would be observed using other types of manipulations targeting uncertainty. In our task, the outcome variance was held constant, making it difficult to isolate the effects of stochasticity on participants' subjective experience of uncertainty. The decision to focus on volatility was based on a rich prior literature demonstrating that volatility manipulations are a reliable means to modulate uncertainty in incremental learning (*Behrens et al., 2007*; *Mathys et al., 2011*; *O'Reilly, 2013*; *Nassar et al., 2012*; *Nassar et al., 2010*; *Browning et al., 2015*; *Piray and Daw, 2020*). Nonetheless, altering outcome variance to capture effects of stochasticity on episodic memory remains a critical avenue for further study. Still other attributes of the learning environment, like valence, have been shown to impact both uncertainty estimation (*Aylward et al., 2019*; *Pulcu and Browning, 2019*) and subsequent memory (*Rosenbaum et al., 2022*; *Kensinger, 2004*). It remains an open question how the valence of outcomes may impact the effects we observed here.

Further, another interpretation of this work is that, rather than capturing a tradeoff between multiple memory systems, our task could possibly be accomplished by a single system learning about, and dynamically weighting, independent features. Specifically, here we operationalized incremental learning as learning about a feature shared across multiple events (deck color) and episodic memory

as learning about a trial-unique feature (an object that could be repeated once). Shifting attention between these independent features whenever one is less reliable could then yield similar behavior to arbitrating between incremental learning and episodic memory as we have posited here. While a scheme like this is possible, much prior work (*Duncan et al., 2019*; *Lee et al., 2015*; *Poldrack et al., 2001*; *Packard and McGaugh, 1996*; *McDonald and White, 1994*; *Wimmer et al., 2014*) indicates that multiple memory systems (differentiated by numerous other behavioral and neural signatures) are involved in the types of repeated vs. one-shot learning measured here. Further, our subsequent memory findings that individual objects and their associated value were better remembered from putatively episodic choices lend further support to the idea that episodic memory is used throughout the task. Nevertheless, more work is needed to distinguish between these alternatives and verify the connection between our task and other signatures of incremental vs. episodic memory.

For example, while in this study we disadvantaged incremental learning relative to episodic memory, similar predictions about their balance could be made by instead preferentially manipulating episodic memory, for example, through effects such as interference or recency and primacy. Another direction would be to look to the computational literature for additional task circumstances in which there are theoretical benefits to deploying episodic memory, and where incremental learning is generally ill suited, such as in environments that are high dimensional or require planning far into the future (*Gershman and Daw, 2017*). In principle, the brain can use episodic memory to precisely target individual past experiences in these situations depending on the relevance of their features to decisions in the present. Recent advances in computational neuroscience have, for example, demonstrated that artificial agents endowed with episodic memory are able to exploit its rich representation of past experience to make faster, more effective decisions (*Lengyel and Dayan, 2007*; *Blundell, 2016*; *Santoro et al., 2016*). While here we provided episodic memory as an alternative source of value to be used in the presence of uncertainty about incremental estimates, future studies making use of paradigms tailored more directly toward episodic memory's assets will help to further elucidate how and when the human brain recruits episodic memory for decisions.

Finally, it is worth noting that many individuals, in both the main and replication samples, failed to meet our baseline performance criterion of altering the incremental learning rate between the low- and high-volatility environments (see 'Materials and methods'). It is unclear whether this insensitivity to volatility was due to the limitations of online data collection, such as inattentiveness, or whether it is a more general feature of human behavior. While the low-volatility environment used here had half as many reversals as the high-volatility environment, it was still much more volatile than some environments used previously to study the effects of volatility on incremental learning (e.g., in entirely stable environments; *Behrens et al., 2007*). Thus, the relatively subtle difference between environments may also have contributed to some participants' volatility insensitivity.

In conclusion, we have demonstrated that uncertainty induced by volatile environments impacts whether incremental learning or episodic memory is recruited for decisions. Greater uncertainty increased the likelihood that single experiences were retrieved for decision-making. This effect suggests that episodic memory aids decision-making when simpler sources of value are less accurate. By focusing on uncertainty, our results shed light on the exact circumstances under which episodic memory is used for decision-making.

## Materials and methods
### Experimental tasks
The primary experimental task used here builds upon a paradigm previously developed by our lab (*Duncan et al., 2019*) to successfully measure the relative contribution of incremental and episodic memory to decisions (*Figure 1A*). Participants were told that they would be playing a card game where their goal was to win as much money as possible. Each trial consisted of a choice between two decks of cards that differed based on their color (shown in *Figure 1* as purple and orange). Participants had 2 s to decide between the decks and, upon making their choice, a green box was displayed around their choice until the full 2 s had passed. The outcome of each decision was then immediately displayed for 1 s. Following each decision, participants were shown a fixation cross during the inter-trial interval period that varied in length (mean = 1.5 s, min = 1 s, max = 2 s). Decks were equally likely

to appear on either side of the screen (left or right) on each trial and screen side was not predictive of outcomes. Participants completed a total of 320 trials and were given a 30 s break every 80 trials.

Participants were made aware that there were two ways they could earn bonus money throughout the task, which allowed for the use of incremental and episodic memory, respectively. First, at any point in the experiment one of the two decks was 'lucky,' meaning that the expected value ($V$) of one deck color was higher than the other ($V_{lucky}$ = 63¢, $V_{unlucky}$ = 37¢). Outcomes ranged from $0 to $1 in increments of 20¢. Critically, the mapping from $V$ to deck color underwent an unsignaled reversal periodically throughout the experiment (*Figure 1B*), which incentivized participants to utilize each deck's recent reward history in order to determine the identity of the currently lucky deck. Each participant completed the task over two environments (with 160 trials in each) that differed in their relative volatility: a low-volatility environment with 8 $V$ reversals, occurring every 20 trials on average, and a high-volatility environment with 16 $V$ reversals, occurring every 10 trials on average. Reversal trials in each environment were determined by generating a list of bout lengths (high volatility: 16 bouts between 6 trials minimum and 14 trials maximum; low volatility: 8 bouts between 15 trials minimum and 24 trials maximum) at the beginning of the task and then randomizing this list for each participant. Participants were told that they would be playing in two different casinos and that in one casino deck luckiness changed less frequently while in the other deck luckiness changed more frequently. Participants were also made aware of which casino they were currently in by a border on the screen, with a solid black line indicating the low-volatility casino and a dashed black line indicating the high-volatility casino. The order in which the environments were seen was counterbalanced across participants.

Second, in order to allow us to assess the use of episodic memory throughout the task, each card within a deck featured an image of a trial-unique object that could reappear once throughout the experiment after initially being chosen. Participants were told that if they encountered a card a second time it would be worth the same amount as when it was first chosen, regardless of whether its deck color was currently lucky or not. On a given trial $t$, cards chosen once from trials $t - 9$ through $t - 30$ had a 60% chance of reappearing following a sampling procedure designed to prevent each deck's expected value from becoming skewed by choice, minimize the correlation between the expected value of previously seen cards and deck expected value, and ensure that choosing a previously selected card remained close to 50¢. Specifically, outcomes for each deck were drawn from a pseudo-random list of deck values that was generated at the start of the task, sampled without replacement, and repopulated after each reversal. Previously seen cards were then sampled using the following procedure: (i) a list of objects from the past 9–30 trials equal to an outcome left in the current list of potential deck outcomes was generated; (ii) the list was narrowed down to objects whose value was incongruent with the current expected value of their associated deck if such objects were available; and (iii) if the average value of objects shown to a participant was greater than 50¢, the object with the lowest value was shown, otherwise an object was randomly sampled without replacement. This sampling procedure is identical to that used previously in *Duncan et al., 2019*.

Participants also completed a separate decision-making task prior to the combined deck learning and card memory task that was identical in design but lacked trial-unique objects on each card. This task, the deck learning task, was designed to isolate the sole contribution of incremental learning to decisions and to allow participants to gain prior experience with each environment's volatility level. In this task, all participants first saw the low-volatility environment followed by the high-volatility environment in order to emphasize the relative increase in the high-volatility environment. Participants completed the combined deck learning and card memory task immediately following completion of the deck learning task and were told that the likelihood of deck luckiness reversals in each environment would be identical for both the deck learning task and the deck learning and card memory task. Instructions were presented immediately prior to each task, and participants completed five practice trials and a comprehension quiz prior to starting each.

Following completion of the combined deck learning and card memory task, we tested participants' memory for the trial-unique objects. Participants completed 80 (up to) three-part memory trials. An object was first displayed on the screen, and participants were asked whether or not they had previously seen the object and were given five response options: Definitely New, Probably New, Don't Know, Probably Old, and Definitely Old. If the participant indicated that they had not seen the object before or did not know, they moved on to the next trial. If, however, they indicated that they had seen the object before, they were then asked if they had chosen the object or not. Lastly, if they

responded that they had chosen the object, they were asked what the value of that object was (with options spanning each of the six possible object values between $0 and $1). Of the 80 trials, 48 were previously seen objects and 32 were new objects that had not been seen before. Of the 48 previously seen objects, half were sampled from each environment (24 each) and, of these, an equal number were taken from each possible object value (with 4 from each value in each environment). As with the decision-making tasks, participants were required to pass a comprehension quiz prior to starting the memory task.

All tasks were programmed using the jsPsych JavaScript library (*de Leeuw, 2015*) and hosted on a Google Cloud server running Apache and the Ubuntu operating system. Object images were selected from publicly available stimulus sets (*Konkle and Oliva, 2012*; *Brady et al., 2008*) for a total of 665 unique objects that could appear in each run of the experiment.

## Participants

A total of 418 participants between the ages of 18–35 were recruited for our main sample through Amazon Mechanical Turk using the Cloud Research Approved Participants feature (*Litman et al., 2017*). Recruitment was restricted to the United States, and $9 compensation was provided following completion of the 50 min experiment. Participants were also paid a bonus in proportion to their final combined earnings on both the training task and the combined deck learning and card memory task (total earnings/100). Before starting each task, all participants were required to score 100% on a quiz that tested their comprehension of the instructions and were made to repeat the instructions until this score was achieved. Informed consent was obtained with approval from the Columbia University Institutional Review Board.

From the initial pool, participants were excluded from analysis on the deck learning and card memory task if they (i) responded to fewer trials than the group average minus 1 standard deviation on the deck learning and card memory task, (ii) responded faster than the group average minus 1 standard deviation on this task, or (iii) did not demonstrate faster learning in the high- compared to the low-volatility environment on the independent deck learning task. Our reasoning for this latter decision was that it is only possible to test for effects of volatility on episodic memory recruitment in participants who were sensitive to the difference in volatility between the environments, and it is well-established that a higher learning rate should be used in more volatile conditions (*Behrens et al., 2007*). Further, our independent assessment of deck learning was designed to avoid issues of selection bias in this procedure. We measured the effect of environment on learning by fitting a mixed-effects logistic regression model to predict if subjects chose the lucky deck up to five trials after a reversal event in the deck learning task. For each subject $s$ and trial $t$, this model predicts the probability that the lucky deck was chosen:

$$p\left(ChooseLucky\right) = \sigma\left(\beta_0 + b_{0,s[t]} + TSinceRev_t \times Env_t\left(\beta_1 + b_{1,s[t]}\right)\right)$$

$$\sigma\left(x\right) = \frac{1}{1+e^{-x}}$$

where $\beta$s are fixed effects, $b$ s are random effects, *TSinceRev* is the trial number coded as distance from a reversal event (1–5), and *Env* is the environment a choice was made in coded as –0.5 and 0.5 for the low- and high-volatility environments, respectively. Participants with positive values of $b_1$ can be said to have chosen the lucky deck more quickly following a reversal in the high- compared to the low-volatility environment, and we included only these participants in the rest of our analyses. A total of 254 participants survived after applying these criteria, with 120 participants failing to respond to the volatility manipulation (criteria iii) and 44 participants responding to too few trials (criteria i) or too quickly (criteria ii).

## Deck learning and card memory task behavioral analysis

For regression models described here as well as those in the following sections, fixed effects are reported in the text as the median of each parameter's marginal posterior distribution alongside 95% credible intervals, which indicate where 95% of the posterior density falls. Parameter values outside of this range are unlikely given the model, data, and priors. Thus, if the range of likely values does not include zero, we conclude that a meaningful effect was observed.

We first analyzed the extent to which previously seen (old) objects were used in the combined deck learning and card memory task by fitting the following mixed-effects regression model to predict whether an old object was chosen:

$$p\left(ChooseOld\right) = \sigma\left(\beta_0 + b_{0,s[t]} + OldVal_t\left(\beta_1 + b_{1,s[t]}\right) + TrueDeckVal_t\left(\beta_2 + b_{2,s[t]}\right)\right)$$

where *OldVal* is the centered value (between –0.5 and 0.5) of an old object. We additionally controlled for the influence of deck value on this analysis by adding a regressor, *TrueDeckVal*, which is the centered true average value of the deck on which each object was shown. Trials not featuring old objects were dropped from this analysis.

We then similarly assessed the extent to which participants engaged in incremental learning overall by looking at the impact of reversals on incremental accuracy directly. To do this, we grouped trials according to their distance from a reversal, up to four trials prior to ($t = -4: -1$), during ($t = 0$), and after ($t = 1: 4$) a reversal occurred. We then dummy coded them to measure their effects on incremental accuracy separately. We also controlled for the influence of old object value in this analysis by including in this regression the coded value of a previously seen object (ranging from 0.5 if the value was \$1 on the lucky deck or \$0 on the lucky deck to –0.5 if the value was \$0 on the lucky deck and \$1 on the unlucky deck), for a total of 18 estimated effects:

$$p\left(ChooseLucky\right) = \sigma\left(T_{-4:\,4}\left(\beta_{1:\,9} + b_{1:\,9,s[t]}\right) + T_{-4:\,4} \times OldVal_t\left(\beta_{10:\,18} + b_{10:\,18,s[t]}\right)\right)$$

To next focus on whether there was an effect of environment on the extent to which the value of old objects was used for decisions, we restricted all further analyses involving old objects to 'incongruent' trials, which were defined as trials on which either the old object was high valued (>50¢) and on the unlucky deck or low valued (<50¢) and on the lucky deck. To better capture participants' beliefs, deck luckiness was determined by the best-fitting incremental learning model (see next section) rather than using the experimenter-controlled ground truth: whichever deck had the higher model-derived value estimate on a given trial was labeled the lucky deck. Our logic in using only incongruent trials was that choices that stray from choosing whichever deck is more valuable should reflect choices that were based on the episodic value for an object. Lastly, we defined our outcome measure of EBCI to equal 1 on trials where the 'correct' episodic response was given (i.e., high-valued objects were chosen and low-valued object were avoided), and 0 on trials where the 'correct' incremental response was given (i.e., the opposite was true). A single mixed-effects logistic regression was then used to assess the possible effects of environment *Env* on EBCI:

$$p\left(EBCI\right) = \sigma\left(\beta_0 + b_{0,s[t]} + EnvNoise_t\beta_1 + Env_t\left(\beta_2 + b_{2,s[t]}\right)\right)$$

where *Env* was coded identically to the above analyses. We included a covariate *EnvNoise* in this analysis to account for the possibility that participants are likely to make noisier incremental value-based decisions in the high-volatility compared to the low-volatility environment, which may contribute to the effects of environment on EBCI. To calculate this index, we fit the following mixed-effects logistic regression model to capture an interaction effect of environment and RB model-estimated deck value (see 'Deck learning computational models' section below) on whether the orange deck was chosen:

$$p\left(ChooseOrange\right) = \sigma\Big(\beta_0 + b_{0,s[t]} + \\ DeckVal_t\left(\beta_1 + b_{1,s[t]}\right) + \\ Env_t\left(\beta_2 + b_{2,s[t]}\right) + \\ DeckVal_t \times Env_t\left(\beta_3 + b_{3,s[t]}\right)\Big)$$

We fit this model only to trials without the presence of a previously seen object in order to achieve a measure of noise specific to incremental learning. Each participant's random effect of the interaction between deck value and environment, $b_3$, was then used as the *EnvNoise* covariate in the logistic regression testing for an effect of environment on EBCI.

To assess the effect of episodic-based choices on reaction time (RT), we used the following mixed-effects linear regression model:

$$RT_t = \beta_0 + b_{0,s[t]} + EBCI_t\left(\beta_1 + b_{1,s[t]}\right) +$$
$$Switch_t\left(\beta_2 + b_{2,s[t]}\right) +$$
$$ChosenVal_t\left(\beta_3 + b_{3,s[t]}\right) +$$
$$RU_t\left(\beta_4 + b_{4,s[t]}\right)$$

where *EBCI* was coded as –0.5 for incremental-based trials and 0.5 for episodic-based trials. We also included covariates to control for three other possible effects on RT. The first, *Switch*, captured possible RT slowing due to exploratory decisions, which in the present task required participants to switch from choosing one deck to the other. This variable was coded as –0.5 if a stay occurred and 0.5 if a switch occurred. The second, *ChosenVal*, captured any effects due to the value of the option that may have guided choice, and was set to be the value of the previously seen object on episodic-based trials and the running average true value on incremental-based trials. Finally, the third, *RU*, captured effects due to possible slowing when choices occurred under conditions of greater uncertainty as estimated by the reduced Bayesian model (see below).

## Deck learning computational models

We next assessed the performance of several computational learning models on our task in order to best capture incremental learning. A detailed description of each model can be found in the 'Supplementary methods.' In brief, these included one model that performed ("Rescorla-Wagner style updating [***Rescorla and Wagner, 1972***]") with both a single (RW1α) and a separate (RW2α) fixed learning rate for each environment, two reduced Bayesian (RB) models (***Nassar et al., 2010***) with both a single (RB1*H*) and a separate hazard rate for each environment (RB1*H*), a contextual inference model (CI), and a Rescorla–Wagner model that learned only a single-value estimate (RW1Q). Models were fit to the deck learning task (see 'Posterior inference' and Appendix 3) and used to generate subject-wise estimates of deck value, and where applicable, uncertainty in the combined deck learning and card memory task.

## Combined choice models

After fitting the above models to the deck learning task, parameter estimates for each subject were then used to generate trial-by-trial time series for deck value and uncertainty (where applicable) throughout performance on the combined deck learning and card memory task. Mixed-effects Bayesian logistic regressions for each deck learning model were then used to capture the effects of multiple memory-based sources of value on incongruent trial choices in this task. For each subject $s$ and trial $t$, these models can be written as

$$p\left(ChooseOrange\right) = \sigma\Big(\beta_0 + b_{0,s[t]} +$$
$$DeckVal_t\left(\beta_1 + b_{1,s[t]}\right) +$$
$$Old_t\left(\beta_2 + b_{2,s[t]}\right) +$$
$$OldVal_t\left(\beta_3 + b_{3,s[t]}\right) +$$
$$OldVal_t \times Env_t\left(\beta_4 + b_{4,s[t]}\right)\Big)$$

where the intercept captures a bias toward choosing either of the decks regardless of outcome, *DeckVal* is the deck value estimated from each model, the effect of *Old* captures a bias toward choosing a previously seen card regardless of its value, and *OldVal* is the coded value of a previously seen object (ranging from 0.5 if the value was $1 on the orange deck or $0 on the purple deck to –0.5 if the value was $0 on the orange deck and $1 on the purple deck). To capture variations in sensitivity to old object value due to volatility (represented here by a categorical environment variable, *Env*, coded as –0.5 for the low- and 0.5 for the high-volatility environment), we also included an interaction term between old object value and environment in each model. An additional seventh regression that also incorporated our hypothesized effect of increased sensitivity to old object value when uncertainty about deck value is higher was also fit. This regression was identical to the others but included an additional interaction effect of uncertainty and old object value: $OldVal_t \times Unc_t\left(\beta_5 + b_{5,s[t]}\right)$ and used the RB2*H* model's

*DeckVal* estimate alongside its estimate of RU to estimate the effect of $OldVal \times Unc$. RU was chosen over CPP because it captures the reducible uncertainty about deck value, which is the quantity we were interested in for this study. Prior to fitting the model, all predictors were z scored in order to report effects in standard units.

## Relative uncertainty analyses

We conducted several other analyses that tested effects on or of RU throughout the combined deck learning and card memory task. RU was mean-centered in each of these analyses. First, we assessed separately the effect of RU at retrieval time on EBCI using a mixed-effects logistic regression:

$$p\left(EBCI\right) = \sigma\left(\beta_0 + b_{0,s[t]} + RU_t\left(\beta_1 + b_{1,s[t]}\right) + RU_t^2\left(\beta_2 + b_{2,s[t]}\right)\right)$$

An additional binomial term was included in this model to allow for the possibility that the effect of RU is nonlinear, although this term was found to have no effect. The effect of RU at encoding time was assessed using an identical model but with RU at encoding included instead of RU at retrieval.

Next, to ensure that the RB model captured uncertainty related to changes in deck luckiness, we tested for an effect of environment on RU using a mixed-effects linear regression:

$$RU_t = \beta_0 + b_{0,s[t]} + Env_t\left(\beta_1 + b_{1,s[t]}\right)$$

We then also looked at the impact of reversals on RU. To do this, we calculated the difference in RU on reversal trials and up to four trials following a reversal from the average RU on the four trials immediately preceding a reversal. Then, using a dummy coded approach similar to that used for the model testing effects of reversals on incremental accuracy, we fit the following mixed-effects linear regression with five effects:

$$RUDifference_t = T_{0:4}\left(\beta_{1:5} + b_{1:5,s[t]}\right)$$

We also assessed the effect of RU on reaction time using another mixed-effects linear regression:

$$RT_t = \beta_0 + b_{0,s[t]} + RU_t\left(\beta_1 + b_{1,s[t]}\right)$$

## Subsequent memory task behavioral analysis

Performance on the subsequent memory task was analyzed in several ways across recognition memory and value memory trials. We first assessed participants' recognition memory accuracy in general by computing the signal detection metric d prime for each participant adjusted for extreme proportions using a log-linear rule (**Hautus, 1995**). The relationship with d prime and sensitivity to both episodic value and incremental value was then determined using simple linear regressions of the form $dprime_s = \beta_0 + Sensitivity_s\left(\beta_1\right)$, where *Sensitivity* was either the random effect of episodic value from the combined choice model for each participant or the random effect of incremental value from the combined choice value for each participant. We additionally assessed the difference in recognition memory performance between environments by computing d prime for each environment separately, with the false alarm rate shared across environments and hit rate differing between environments, using the following mixed-effects linear regression:

$$dprime = \beta_0 + b_{0,s} + Env\left(\beta_1 + b_{1,s}\right)$$

We next determined the extent to which participants' memory for previously seen objects was impacted by whether an object was seen initially on either an episodic- or incremental-based choice using the following mixed-effects logistic regression model:

$$p\left(Hit_t\right) = \sigma\left(\beta_0 + b_{0,s[t]} + EBCI_t\left(\beta_1 + b_{1,s[t]}\right)\right)$$

where *Hit* was 0 if an object was incorrectly labeled as new and 1 if it was accurately identified as old. The final recognition memory analysis we performed was focused on assessing the impact of variables (RU, changepoint probability [CPP], and the absolute value of prediction error [APE])

extracted from the RB model at encoding time on future subsequent memory. Because these variables are, by definition, highly correlated with one another (see 'Supplementary methods'), we fit separate simple mixed-effects logistic regression models predicting recognition memory from each variable separately and then compared the predictive performance of each model (see below) to determine which best accounted for subsequent memory performance. The models additionally controlled for potential recognition memory enhancements due to the absolute magnitude of an object's true value by including this quantity as a covariate in each of these models.

In addition to the analyses of recognition memory, analogous effects were assessed for performance on memory for value. General value memory accuracy and a potential effect of environment on remembered value were assessed using the following mixed-effect linear regression:

$$
\begin{aligned}
Value_t = \quad & \beta_0 + b_{0,s[t]} + \\
& TrueVal_t \left( \beta_1 + b_{1,s[t]} \right) + \\
& Env_t \left( \beta_2 + b_{2,s[t]} \right) + \\
& Env_t \times TrueVal_t \left( \beta_3 + b_{3,s[t]} \right)
\end{aligned}
$$

where *Value* is the remembered value of an object on each memory trial (between \$0 and \$1), and *TrueVal* is an object's true value. We next assessed whether value memory was similarly impacted by whether an object was seen initially on either ran episodic- or incremental-based choice using a similar model for objects from incongruent trials only with *EBCI* as a predictor rather than *Env*. Lastly, as with the recognition memory analyses, we determined the extent to which trial-wise variables from the RB model (RU, CPP, and APE) at encoding impacted subsequent value memory by using each of these as a predictor instead in similar models and then comparing the predictive performance of each in an identical manner to the recognition memory models.

## Posterior inference and model comparison

Parameters for all incremental learning models were estimated using hierarchical Bayesian inference such that group-level priors were used to regularize subject-level estimates. This approach to fitting reinforcement learning models improves parameter identifiability and predictive accuracy (*van Geen and Gerraty, 2021*). The joint posterior was approximated using No-U-Turn Sampling (*Hoffman and Gelman, 2011*) as implemented in Stan (*Team SD, 2020*). Four chains with 2000 samples (1000 discarded as burn-in) were run for a total of 4000 posterior samples per model. Chain convergence was determined by ensuring that the Gelman–Rubin statistic $R$ was close to 1. A full description of the parameterization and choice of priors for each model can be found in Appendix 3. All regression models were fit using No-U-Turn Sampling in Stan with the same number of chains and samples. Default weakly informative priors implemented in the rstanarm package (*Rstanarm, 2022*) were used for each regression model. Model fit for the combined choice models and the models measuring trial-wise effects of encoding on subsequent memory was assessed by separating each dataset into 20-folds and performing a cross-validation procedure by leaving out N/20 subjects per fold, where N is the number of subjects in each sample. The expected log pointwise predictive density (ELPD) was then computed and used as a measure of out-of-sample predictive fit for each model.

## Replication

We identically repeated all procedures and analyses applied to the main sample on an independently collected replication sample. A total of 401 participants were again recruited through Amazon Mechanical Turk, and 223 survived exclusion procedures carried out identically to those used for the main sample, with 124 participants failing to respond to the volatility manipulation (criteria iii) and 54 participants responding to too few trials (criteria i) or too quickly (criteria ii).

## Citation race and gender diversity statement

The gender balance of papers cited within this work was quantified using databases that store the probability of a first name being carried by a woman. Excluding self-citations to the first and last authors of this article, the gender breakdown of our references is 12.16% woman (first)/woman (last), 6.76% man/woman, 23.44% woman/man, and 57.64% man/man. This method is limited in that (i) names, pronouns, and social media profiles used to construct the databases may not, in every case, be

indicative of gender identity and (b) it cannot account for intersex, nonbinary, or transgender people. Second, we obtained predicted racial/ethnic category of the first and last authors of each reference using databases that store the probability of a first and last name being carried by an author of color. By this measure (and excluding self-citations), our references contain 9.55% author of color (first)/ author of color(last), 19.97% white author/author of color, 22.7% author of color/white author, and 47.78% white author/white author. This method is limited in that (i) using names and Florida Voter Data to make the predictions may not be indicative of racial/ethnic identity, and (ii) it cannot account for indigenous and mixed-race authors, or those who may face differential biases due to the ambiguous racialization or ethnicization of their names.

## Acknowledgements

The authors thank Sam Gershman, Raphael Gerraty, Camilla van Geen, Mariam Aly, and members of the Shohamy Lab for insightful discussion and conversations. Support was provided by the NSF Graduate Research Fellowship (JN; award # 1644869), the NSF (DS, ND; award # 1822619), the NIMH/NIH (DS, ND; award # MH121093), and the Templeton Foundation (DS grant #60844).

## Additional information

### Funding

| Funder | Grant reference number | Author |
| --- | --- | --- |
| National Science Foundation | 1644869 | Jonathan Nicholas |
| National Science Foundation | 1822619 | Nathaniel D Daw<br>Daphna Shohamy |
| National Institutes of Health | MH121093 | Nathaniel D Daw<br>Daphna Shohamy |
| John Templeton Foundation | 60844 | Daphna Shohamy |

The funders had no role in study design, data collection and interpretation, or the decision to submit the work for publication.

### Author contributions

Jonathan Nicholas, Conceptualization, Data curation, Software, Formal analysis, Validation, Investigation, Visualization, Methodology, Writing - original draft, Project administration, Writing – review and editing; Nathaniel D Daw, Conceptualization, Supervision, Methodology, Writing – review and editing; Daphna Shohamy, Conceptualization, Resources, Supervision, Funding acquisition, Methodology, Writing – review and editing

### Author ORCIDs

Jonathan Nicholas ⓘ http://orcid.org/0000-0002-2314-0765
Nathaniel D Daw ⓘ http://orcid.org/0000-0001-5029-1430

### Ethics

Informed consent was obtained online with approval from the Columbia University Institutional Review Board (IRB #1488).

### Decision letter and Author response

Decision letter https://doi.org/10.7554/eLife.81679.sa1
Author response https://doi.org/10.7554/eLife.81679.sa2

# Additional files

## Supplementary files
• MDAR checklist

## Data availability
All code, data, and software needed to reproduce the manuscript can be found here: https://code-ocean.com/capsule/2024716/tree/v1; DOI: https://doi.org/10.24433/CO.1266819.v1.

The following dataset was generated:

| Author(s) | Year | Dataset title | Dataset URL | Database and Identifier |
|---|---|---|---|---|
| Nicholas J, Daw ND, Shohamy D | 2022 | Uncertainty alters the balance between incremental learning and episodic memory | https://doi.org/10.24433/CO.1266819.v1 | Code Ocean, 10.24433/CO.1266819.v1 |

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

# Appendix 1

## Replication results

Here, we repeat and describe all analyses reported in the main text with replication sample. All results are reported in the same order as in the main text.

## Episodic memory is used more under conditions of greater volatility

Participants in the replication sample were substantially more likely to choose high-valued old objects compared to low-valued old objects ($\beta_{OldValue} = 0.723, 95\% CI = [0.624, 0.827]$; *Figure 2—figure supplement 1A*). Participants also altered their behavior in response to reversals in deck value. The higher-valued (lucky) deck was chosen more frequently on trials immediately preceding a reversal ($\beta_{t-4} = 0.095, 95\% \ CI = [0.016, 0.176]$; $\beta_{t-3} = 0.128, 95\% \ CI = [0.047, 0.213]$; $\beta_{t-2} = 0.168, 95\% \ CI = [0.085, 0.251]$; $\beta_{t-1} = 0.161, 95\% CI = [0.075, 0.25]$; *Figure 2—figure supplement 1B*). This tendency was then disrupted by trials on which a reversal occurred ($\beta_{t=0} = -0.373, 95\% CI = [-0.464, -0.286]$), with performance quickly recovering as the newly lucky deck became chosen more frequently on the trials following a reversal ($\beta_{t+1} = -0.256, 95\% CI = [-0.337, -0.175]$; $\beta_{t+2} = -0.144, 95\% CI = [-0.22, -0.064]$; $t + 3$: $\beta_{t+3} = -0.024, 95\% CI = [-0.102, 0.053]$; $\beta_{t+4} = 0.113, 95\% CI = [0.055, 0.174]$). Thus, participants in the replication sample were also sensitive to reversals in deck value, thereby indicating that they engaged in incremental learning throughout the task.

Participants in the replication sample also based more decisions on episodic value in the high-volatility environment compared to the low-volatility environment ($\beta_{Env} = 0.146, 95\% CI = [0.06, 0.228]$; *Figure 2—figure supplement 1C*). Furthermore, decisions based on episodic value again took longer ($\beta_{EBCI} = 39.445, 95\% CI = [29.660, 49.328]$; *Figure 2—figure supplement 1D*).

## Uncertainty increases sensitivity to episodic value

In the replication sample, the reduced Bayesian model with two hazard rates was again the best-fitting model (*Figure 3—figure supplement 1A*). Participants detected higher levels of volatility in the high- compared to the low-volatility environment, as indicated by the generally larger hazard rates recovered from the high- compared to the low-volatility environment ($\beta_{Low} = 0.048, 95\% CI = [0.038, 0.06]$; $\beta_{High} = 0.071, 95\% CI = [0.058, 0.088]$; *Figure 3—figure supplement 1B*). Compared to an average of the four trials prior to a reversal, RU also increased immediately following a reversal and stabilized over time ($\beta_{t=0} = 0.021, 95\% \ CI = [-0.014, 0.056]$; $\beta_{t+1} = -0.22, 95\% \ CI = [-0.253, -0.185]$; $\beta_{t+2} = -0.144, 95\% \ CI = [-0.178, -0.11]$; $\beta_{t+3} = -0.098, 95\% \ CI = [-0.129, -0.064]$; $\beta_{t+4} = -0.05, 95\% \ CI = [-0.083, -0.019]$; *Figure 3—figure supplement 1C*). RU was again also, on average, greater in the high- compared to the low-volatility environment ($\beta_{Env} = 0.01, 95\% CI = [0.007, 0.013]$) and related to reaction time such that choices made under more uncertain conditions took longer ($\beta_{RU} = 1.364, 95\% CI = [0.407, 2.338]$).

Episodic memory was also used more on incongruent trial decisions made under conditions of high RU ($\beta_{RU} = 2.718, 95\% CI = [1.096, 4.436]$; *Figure 4—figure supplement 1A*). We again fit the combined choice model to the replication sample and found the following. Participants again used both sources of value throughout the task: both deck value as estimated by the model ($\beta_{DeckValue} = 0.431, 95\% CI = [0.335, 0.516]$; *Figure 4—figure supplement 1B*) and the episodic value from old objects ($\beta_{OldValue} = 0.191, 95\% CI = [0.137, 0.245]$) strongly impacted choice. Lastly, episodic value again impacted choices more when RU was high ($\beta_{OldValue:RU} = 0.043, 95\% CI = [0.00003, 0.088]$) and in the high- compared to the low-volatility environment ($\beta_{OldValue:Env} = 0.092, 95\% CI = [0.047, 0.136]$).

Finally, there was again no relationship between the use of episodic memory on incongruent trial decisions and RU at encoding ($\beta_{RU} = 0.99, 95\% CI = [-0.642, 2.576]$; *Figure 4—figure supplement 2*). Including a sixth parameter to assess increased sensitivity to old object value due to RU at encoding time did not have an effect in the combined choice model ($\beta_{OldValue:RU} = -0.003, 95\% CI = [-0.046, 0.037]$; *Figure 4—figure supplement 2*), which is also reported in the main text. As with the main sample, including this parameter did not provide a better fit to subjects' choices than the combined choice model with only increased sensitivity due to RU at retrieval time.

## Episodic and incremental value sensitivity predicts subsequent memory performance

Participants in the replication sample again performed well above chance on the test of recognition memory ($\beta_0 = 1.874, 95\% CI = [1.772, 1.977]$), and objects from episodic choice trials were better

remembered than those from incremental choice trials ($\beta_{EBCI} = 0.157, 95\%CI = [0.033, 0.278]$; *Figure 5—figure supplement 1A*). Recall for the value of previously seen objects was also well predicted by their true value ($\beta_{TrueValue} = 0.181, 95\%CI = [0.162, 0.120]$) and value recall was improved for objects from episodic choice trials ($\beta_{EBCI: TrueValue} = 0.049, 95\%CI = [0.030, 0.067]$; *Figure 5—figure supplement 1B*). Participants with better subsequent recognition memory were again more sensitive to episodic value ($\beta_{EpSensitivity} = 0.334, 95\%CI = [0.229, 0.44]$; *Figure 5—figure supplement 1C*), and these same participants were again less sensitive to incremental value ($\beta_{IncSensitivity} = -0.124, 95\%CI = [-0.238, -0.009]$; *Figure 5—figure supplement 1D*).

## Appendix 2

### Uncertainty during encoding improves subsequent memory in both samples

The subsequent memory task provided us with the opportunity to test whether participants have better subsequent memory for objects encoded under conditions of greater uncertainty. Supporting the notion that uncertainty improves subsequent memory, recognition memory for objects encoded in the high-volatility environment was better than for those encoded in the low-volatility environment (main: $\beta_{Env} = 0.053, 95\%CI = [0.009, 0.098]$; replication: $\beta_{Env} = 0.078, 95\%CI = [0.031, 0.126]$). This coarse effect was limited to recognition memory, however, as memory for object value was less impacted by the environment in which it was seen (main: $\beta_{Env} = -0.002, 95\%CI = [-0.012, 0.009]$; replication: $\beta_{Env} = 0.008, 95\%CI = [-0.002, 0.019]$).

We next examined the impact of RU at encoding on subsequent memory. Both recognition memory (main: $\beta_{RU} = 0.129, 95\%CI = [0.022, 0.241]$; replication: $\beta_{RU} = 0.179, 95\%CI = [0.041, 0.329]$) and value memory (main: $\beta_{TrueValue:RU} = 0.012, 95\%CI = [0.001, 0.023]$; replication: $\beta_{TrueValue:RU} = 0.012, 95\%CI = [0.001, 0.023]$; ***Figure 5—figure supplement 2***) were associated with greater RU at encoding time. Lastly, we assessed how these effects of uncertainty at encoding compared to the effects of surprise, which is thought to also improve subsequent memory and is separately estimated by the RB model (see 'Supplementary methods'). We found that surprise at encoding (quantified here as both the probability of a reversal in deck value and the absolute value of reward prediction error) led to modest improvement in subsequent memory, but these effects were less consistent across samples and types of memory (***Figure 5—figure supplement 2***). Models of subsequent memory performance featuring surprise were also outperformed by those that instead predicted memory from RU. Together, these results indicate that the presence of uncertainty at encoding improves subsequent memory.

## Appendix 3

### Supplementary methods

#### Description of incremental learning models

##### Rescorla–Wagner (RW)

The first model we considered was a standard model-free reinforcement learner that assumes a stored value ($Q$) for each deck is updated over time. $Q$ is then referenced on each decision in order to guide choices. After each outcome $o_t$, the value for the orange deck $Q_O$ is updated according to the following rule (*Rescorla and Wagner, 1972*) if the orange deck is chosen:

$$Q_{O,t+1} = Q_{O,t} + \alpha \left( o_t - Q_{O,t} \right)$$

And is not updated if the purple deck is chosen:

$$Q_{O,t+1} = Q_{O,t}$$

Likewise, the value for the purple deck $Q_B$ is updated equivalently. Large differences between estimated value and outcomes therefore have a larger impact on updates, but the overall degree of updating is controlled by the learning rate, $\alpha$. Two versions of this model were fit, one with a single learning rate (RW1α), and one with two learning rates (RW2α), $\alpha_{low}$ or $\alpha_{high}$, depending on which environment the current trial was completed in. These parameters are constrained to lie between 0 and 1. A separate learning rate was used for each environment in the (RW2α) version to capture the well-established idea that a higher learning rate should be used in more volatile conditions (*Behrens et al., 2007*). A third RW model (RW1Q), also with two learning rates, was additionally fit to better match the property of the reduced Bayesian model (described below) in which anticorrelation between each deck's value is assumed due to learning only a single value. This was accomplished by forcing the model to learn only one $Q$, where outcomes were coded in terms of the orange deck. For example, this means that an outcome worth \$1 on the orange deck is treated the same as an outcome worth \$0 on the purple deck by this model.

##### Reduced Bayesian (RB)

The second model we considered was the reduced Bayesian (RB) model developed by Nassar and colleagues (*Nassar et al., 2010*). This model tracks and updates its belief that the orange deck is lucky based on trial-wise outcomes, $o_t$, using the following prediction error-based update:

$$B_{t+1} = B_t + \alpha_t \left( o_t - B_t \right)$$

This update is identical to that used in the RW model; however, the learning rate $\alpha_t$ is itself updated following each outcome according to the following rule:

$$\alpha_t = \Omega_t + \left( 1 - \Omega_t \right) \tau_t$$

where $\Omega_t$ is the probability that a change in deck luckiness has occurred on the most recent trial (the CPP) and $\tau_t$ is the imprecision in the model's belief about deck value (the RU). The learning rate therefore increases whenever CPP or RU increases. CPP can be written as

$$\Omega_t = \frac{\mathcal{U} \left( o_t | 0, 1 \right) H}{\mathcal{U} \left( o_t | 0, 1 \right) H + \mathcal{N} \left( o_t | B_t, \sigma^2 \right) \left( 1 - H \right)}$$

where $H$ is the hazard rate or probability of a change in deck luckiness. Two versions of this model were fit, one with a single hazard rate (RB1$H$), and one with two hazard rates (RB2$H$), $H_{low}$ and $H_{high}$, depending on the environment the current trial was completed in. In this equation, the numerator represents the probability that an outcome was sampled from a new average deck value, whereas the denominator indicates the combined probability of a change and the probability that the outcome was generated by a Gaussian distribution centered around the most recent belief about deck luckiness and the variance of this distribution, $\sigma^2$. Because CPP is a probability, it is constrained to lie between 0 and 1. In our implementation, $H$ was a free parameter (see 'Posterior inference' section below) and $\Omega_1$ was initialized to 1.

RU, which is the uncertainty about deck value relative to the amount of noise in the environment, is quite similar to the Kalman gain used in Kalman filtering:

$$k_t = \Omega_t \sigma^2 + (1 - \Omega_t) \tau_t \sigma^2 + \Omega_t (1 - \Omega_t) ((o_t - B_t)(1 - \tau_t))^2$$

$$\tau_{t+1} = \frac{k_t}{k_t + \sigma^2}$$

where $\sigma^2$ is the observation noise and was here fixed to the true observation noise (0.33). $k_t$ consists of three terms: the first is the variance of the deck value distribution conditional on a change point, the second is the variance of the deck value distribution conditional on no change, and the third is the variance due to the difference in means between these two distributions. These terms are then used in the equation for $\tau_{t+1}$ to provide the uncertainty about whether an outcome was due to a change in deck value or the noise in observations that is expected when a change point has not occurred. Because this model does not follow the two-armed bandit assumption of our task (i.e., that outcomes come from two separate decks), all outcomes were coded in terms of the orange deck, as in the RW1Q model described above. While this description represents a brief overview of the critical equations of the reduced Bayesian model, a full explanation can be found in **Nassar et al., 2010**.

## Softmax choice

All incremental learning models were paired with a softmax choice function in order to predict participants' decisions on each trial:

$$\theta_t = \frac{1}{1 + e^{-(\beta_0 + \beta_1 V_t)}}$$

where $\theta_t$ is the probability that the orange deck was chosen on trial $t$. This function also consists of two inverse temperature parameters: $\beta_0$ to model an intercept and $\beta_1$ to model the slope of the decision function related to deck value. The primary difference for each model was how $V_t$ is computed: RW ($V_t = Q_{O,t} - Q_{B,t}$); RB ($V_t = B_t$); RW1Q ($V_t = Q_t$). In each of these cases, a positive $V_t$ indicates evidence that the orange deck is more valuable while a negative $V_t$ indicates evidence that the purple deck is more valuable.

## Posterior inference

For all incremental learning models, the likelihood function can be written as

$$c_{s,t} \sim Bernoulli\left(\theta_{s,t}\right)$$

where $c_{s,t}$ is 1 if subject $s$ chose the orange deck on trial $t$ and 0 if purple was chosen. Following the recommendations of **Gelman and Hill, 2006** and **van Geen and Gerraty, 2021**, $\beta_s$ is drawn from a multivariate normal distribution with mean vector $\mu_\beta$ and covariance matrix $\Sigma_\beta$:

$$\beta_s \sim MultivariateNormal\left(\mu_\beta, \Sigma_\beta\right)$$

where $\Sigma_\beta$ is decomposed into a vector of coefficient scales $\tau_\beta$ and a correlation matrix $\Omega_\beta$ via

$$\Sigma_\beta = diag\left(\tau_\beta\right) \times \Omega_\beta \times diag\left(\tau_\beta\right)$$

Weakly informative hyperpriors were then set on the hyperparameters $\mu_\beta, \Omega_\beta$, and $\tau_\beta$:

$$\mu_\beta \sim \mathcal{N}\left(0, 5\right)$$

$$\tau_\beta \sim Cauchy^+\left(0, 2.5\right)$$

$$\Omega_\beta \sim LKJCorr\left(2\right)$$

These hyperpriors were chosen for their respective desirable properties: the half Cauchy is bounded at zero and has a relatively heavy tail that is useful for scale parameters, the LKJ prior

with shape = 2 concentrates some mass around the unit matrix, thereby favoring less correlation (**Lewandowski et al., 2009**), and the normal is a standard choice for regression coefficients.

Because sampling from heavy-tailed distributions like the Cauchy is difficult for Hamiltonian Monte Carlo (**Team SD, 2020**), a reparameterization of the Cauchy distribution was used here. $\tau_\beta$ was thereby defined as the transform of a uniformly distributed variable $\tau_{\beta\_}u$ using the Cauchy inverse cumulative distribution function such that

$$F_x^{-1}\left(\tau_{\beta\_}u\right) = \tau_\beta\left(\pi\left(\tau_{\beta\_}u - \tfrac{1}{2}\right)\right)$$

$$\tau_{\beta\_}u \sim \mathcal{U}\left(0,1\right)$$

In addition, a multivariate noncentered parameterization specifying the model in terms of the Cholesky factorized correlation matrix was used in order to shift the data's correlation with the parameters to the hyperparameters, which increases the efficiency of sampling the parameters of hierarchical models (**Team SD, 2020**). The full correlation matrix $\Omega_\beta$ was replaced with a Cholesky factorized parameter $L_{\Omega_\beta}$ such that

$$\Omega_\beta = L_{\Omega_\beta} \times L_{\Omega_\beta}^T$$

$$\beta_s = \mu_\beta + \left(diag\left(\tau\right) \times L_{\Omega_\beta} \times z\right)^T$$

$$L_{\Omega_\beta} \sim LKJCholesky\left(2\right)$$

$$z \sim \mathcal{N}\left(0,1\right)$$

where multiplying the Cholesky factor of the correlation matrix by the standard normally distributed additional parameter $z$ and adding the group mean $\mu_\beta$ creates a $\beta_s$ vector distributed identically to the original model.

While the choice function is identical for each model, the parameters used in generating deck value differ for each. All were fit hierarchically and were modeled with the following priors and hyperpriors:

Rescorla–Wagner with a single learning rate (RW1α):

$$\alpha \sim \beta\left(a1, a2\right)$$
$$a1 \sim \mathcal{N}\left(0,5\right)$$
$$a2 \sim \mathcal{N}\left(0,5\right)$$

Rescorla–Wagner with two learning rates (RW2α) and with one Q-value (RW1Q):

$$\alpha_{low} \sim \beta\left(a1_{low}, a2_{low}\right)$$
$$\alpha_{high} \sim \beta\left(a1_{high}, a2_{high}\right)$$
$$a1_{low} \sim \mathcal{N}\left(0,5\right)$$
$$a2_{low} \sim \mathcal{N}\left(0,5\right)$$
$$a1_{high} \sim \mathcal{N}\left(0,5\right)$$
$$a2_{high} \sim \mathcal{N}\left(0,5\right)$$

Reduced Bayes with a single hazard rate (RB1H):

$$H \sim \beta\left(h1, h2\right)$$
$$h1 \sim \mathcal{N}\left(0,5\right)$$
$$h2 \sim \mathcal{N}\left(0,5\right)$$

Reduced Bayes with two hazard rates (RB2H):

$$H_{low} \sim \beta\left(h1_{low}, h2_{low}\right)$$
$$H_{high} \sim \beta\left(h1_{high}, h2_{high}\right)$$
$$h1_{low} \sim \mathcal{N}\left(0,5\right)$$
$$h2_{low} \sim \mathcal{N}\left(0,5\right)$$
$$h1_{high} \sim \mathcal{N}\left(0,5\right)$$
$$h2_{high} \sim \mathcal{N}\left(0,5\right)$$

## Description of contextual inference model

Because of the structure of our task, one possibility is that participants did not engage in incremental learning, but instead inferred which one of two switching contexts they were in (either that the orange deck was lucky and the purple deck was unlucky or vice versa). To address this, we developed a contextual inference (CI) model based on a standard hidden Markov model (HMM) with two latent states. While HMMs are covered extensively elsewhere (*Rabiner and Juang, 1986*), we provide the following brief overview. The model assumes that each outcome, $o_t$, was generated by a hidden state, $s_t$, which may take one of two values on each trial, $s_t \in [1,2]$. The goal of the model is then to infer which of the two states gave rise to each outcome on each trial using the following generative model:

$$o_t \sim \mathcal{N}\left(\mu_s, 1\right)$$
$$s_t \sim Categorical\left(\theta_{s_{t-1}}\right)$$

where $\mu \in [1,2]$, and $\theta$ is a 2 × 2 transition matrix. Here, we assume that each outcome is normally distributed with a known scale parameter and unknown location parameters, $(\mu_1, \mu_2)$. The state variable follows a categorical distribution parameterized by $\theta$, which determines the likelihood that, on a given trial, each state will transition to either the other state or itself. Here, $\theta$ was modeled separately for each environment to mirror the difference in volatility between environments. μ and θ were then fit as free parameters for each participant using Hamiltonian Monte Carlo, following recommendations for fitting HMMs in Stan (*Team SD, 2020*). The following priors were used for each parameter:

$$\theta_{low} \sim Dirichlet\left(1,1\right)$$
$$\theta_{high} \sim Dirichlet\left(1,1\right)$$
$$\mu_1 \sim \mathcal{N}\left(V_{lucky}, \sigma\right)$$
$$\mu_2 \sim \mathcal{N}\left(V_{unlucky}, \sigma\right)$$

where $\sigma$ is the true standard deviation of outcomes, and $V_{lucky}$ and $V_{unlucky}$ are the true expected values of the lucky and unlucky decks, respectively.

We then calculated the likelihood of each participant's sequence of outcomes using the forward algorithm to compute the following marginalization:

$$p\left(o|\theta, \mu\right) = \sum_s p\left(o, s|\theta, \mu\right)$$

Upon estimating the parameters, the most probable sequence of states to have generated the observed outcomes was computed using the Viterbi algorithm. Assigning a state to each timepoint allowed us to make use of the assigned state's μ as the expected state value for the timepoint. This was then treated as the deck value for further analyses, as for the incremental learning models. Lastly, outcomes were coded similarly to the RB and RW1Q models.

