## [Editor Report]

This paper posits that higher uncertainty environments should lead to more reliance on episodic memory, finding compelling evidence for this idea across several analysis approaches and across two independent samples. This is an important paper that will be of interest to a broad group of learning, memory, and decision-making researchers.

---

## [Decision Letter]

**Decision letter after peer review:**

Thank you for submitting your article "Uncertainty alters the balance between incremental learning and episodic memory" for consideration by *eLife*. Your article has been reviewed by 3 peer reviewers, one of whom is a member of our Board of Reviewing Editors, and the evaluation has been overseen by Michael Frank as the Senior Editor. The reviewers have opted to remain anonymous.

All of the reviewers felt that this was a promising paper with compelling results, but they also raised a number of questions about the methodology and interpretation of the results that should be addressed in a revision, as detailed below.

*Reviewer #1 (Recommendations for the authors):*

The wide vs. skinny error bars are very difficult to visually differentiate. I recommend a more obvious difference.

*Reviewer #2 (Recommendations for the authors):*

1. 40-45% of the participants are excluded from the analysis in the main and replication samples. The authors should clarify how many were excluded for each criterion. Was the main culprit whether people responded to the volatility manipulation in the original deck learning task? If so, what does this mean about the generality of the effects of uncertainty in incremental learning? I suspect this may be due to the relative attentiveness of online participants, but the authors should address the caveat in the text. Some aspects of these excluded data may still be relevant. For example, if these participants do not register any differences in uncertainty between the two environments, wouldn't the prediction be that their use of episodic memory also does not differ?

2. Some aspects of the methods were not clear.

a. Was the order of the high and low volatility blocks counterbalanced across participants? Was the order the same in the first deck learning and second deck learning + card memory tasks? Were participants told explicitly that the environments would carry over between the two tasks? (The last point would further support using estimates fit to the first task out of the sample in the second.)

b. How individual objects were re-sampled was described in the overview (lines 435-439), but I imagine the details will matter a lot to people interested in replicating and extending this work. It would help to point the interested reader to where these could be found (eg, is the code provided, or is it based on a previous study described in detail, etc.).

3. Given that the results precede the methods, there are some aspects of the task that would be helpful to explain at the outset (around Figure 1). I was initially confused because the outcome in Figure 1 is $1 but the value memory was on a scale of 0-100, but this could be cleared up with a sentence about the possible outcomes. It would also be helpful to mention the mean outcome on the lucky versus unlucky deck, how frequently the lucky deck changes, and what participants are told explicitly about the volatility manipulation. This could be done in the text or with a revision to Figure 1. I was also confused about how the two samples were used until the end of the Results section (are they being analyzed together or separately? What am I looking at in Figures 2-5?). Again, a well-placed sentence at the top of the Results section would clear this up.

4. In Figure 3, I think a slightly different comparison would be useful, in addition to or instead of the two Rescorla-Wagner models. One difference between the reduced Bayesian and RW models is that the learning rate is dynamic in the RB model but not the RW model. But another difference between the way the two models are implemented is that the RB model assumes the value of the two decks are perfectly anti-correlated (ie, it is learning only one value estimate), while the RW model does not (ie, it is learning about the two decks independently). Thus, the RB model assumes a critical aspect of the structure of the task that the RW model does not. I doubt this difference completely accounts for its better performance, but this should be tested. A δ-rule model with a fixed learning rate that learns a single value estimate (like the RB model) would be the needed comparison. This comparison would also isolate the effect of including the dynamic learning rate (according to RU and CPP) in the model.

5. The discussion goes into the different effects of novelty, surprise, and uncertainty on subsequent memory (lines 349-364), in the context of the lack of effect of uncertainty (RU from the reduced Bayesian model) at encoding. But have the authors looked at the effect of surprise (changepoint probability in the reduced Bayesian model) at encoding? The previous studies discussed here would predict that surprise at encoding should enhance subsequent memory (and perhaps the use of episodic memory in choice). This point is not central to the manuscript, of course, but the authors have additional data relevant to the distinctions they are raising here.

*Reviewer #3 (Recommendations for the authors):*

As mentioned in the public review, I thought this was a very interesting study and the results were clearly communicated. However, I have a number of questions/recommendations for the authors to strengthen the results and interpretation.

Regarding the points made in the public review about uncertainty v volatility and context, I'd make the following recommendations:

(1) Uncertainty vs. volatility (described in public review): it would make sense to reframe results around volatility rather than uncertainty, or strengthen the trial-wise RU analyses to look at trial-wise RU only at trials far away from reversals. At least there should be some discussion of where uncertainty arises besides volatility.

(2) Context: I think the analyses would be strengthened with an additional model using context inference rather than incremental learning. A natural choice might be Gershman and Niv 2012, although you could possibly get away with something simpler if you assume 2 contexts.

(3) The focus on incongruent trials seems potentially thorny. It is intuitive why the authors do this: trials in which episodic and incremental values disagree are informative about which normative strategy the subjects are using. However, in the high volatility condition, if subjects are using incremental value, they may be more likely to have an outdated incremental value which would look consistent with the episodic choice. I would propose the authors look at congruent trials as well to confirm that they are indeed less likely to make errors on these trials in the high volatility condition than they are in the low volatility condition.

(4) Another question relates to the interpretation of competing for episodic v. incremental strategies, as opposed to just learning about independent features. One could argue that the subjects are doing instrumental learning over objects and colors separately, and when the reliability of one feature (color) is decremented, the other feature is relatively up-weighted. This also seems consistent with the fact that episodic and incremental learning tradeoff -- attending more to the object feature would perhaps compete with color.

(5) The authors show that uncertainty at encoding time does not have a discernible effect on the episodic index. This is evidence that volatility is modulating episodic contribution to decision-making, rather than encoding strength, which is a pretty fundamental part of the results (and in contrast to eg Sun et al. 2021, where unpredictability modulates episodic consolidation). One key thing to look at is if subjects show any difference in their ability to recall familiarity/value of objects from the different conditions. This would also speak to the question of if volatility is affecting encoding rather than just recall during decision-making. It would also make sense to look at the role of other variables at encoding time (eg prediction error) to see if these predict future use. It would also be interesting to see if subjects are storing the object value or the incremental value at the time the object was first shown -- this would be easy to check (when subjects rate the value in the last block, are they more likely to err in the direction of the incremental value at the time of encoding (eg like Figure 2A but with x-axis = incremental value at the time estimated with RW)). This would shed insight into exactly what kind of episodic strategy the subjects are deploying.

(6) The analysis showing that subjects that were better at recall in block 3 also had higher episodic index was a useful sanity check. It seems it would also be possible to perform this analysis within-subject (eg does episodic choice correlate with accurate value memory) and that would bear more on the question of whether it was uncertainty or simply a subjective preference for one strategy or another.

---

## [Author Response]

Reviewer #1 (Recommendations for the authors):The wide vs. skinny error bars are very difficult to visually differentiate. I recommend a more obvious difference.

Thank you. We agree that it was difficult to differentiate between the 80% and 95% posterior intervals that were plotted around group-level estimates. This is largely because there was little difference between these intervals on the scale that we are plotting in order to visualize individual subject estimates in addition to group-level estimates. In the revision, we have altered figures that previously had both intervals (e.g. Figure 3B and Figure 4B) to have only 95% posterior intervals, as these are more informative and in line with what is reported throughout the rest of the paper. We have additionally changed the visualization of group-level estimates in Figure 4B from lines to bars in order to more explicitly differentiate between how error and estimates are visualized.

Reviewer #2 (Recommendations for the authors):1. 40-45% of the participants are excluded from the analysis in the main and replication samples. The authors should clarify how many were excluded for each criterion. Was the main culprit whether people responded to the volatility manipulation in the original deck learning task? If so, what does this mean about the generality of the effects of uncertainty in incremental learning? I suspect this may be due to the relative attentiveness of online participants, but the authors should address the caveat in the text. Some aspects of these excluded data may still be relevant. For example, if these participants do not register any differences in uncertainty between the two environments, wouldn't the prediction be that their use of episodic memory also does not differ?

Thank you for giving us the opportunity to clarify. We have now added how many participants were excluded due to insensitivity to the volatility manipulation or for their general performance during the second task in the text (lines 603-605 and 786-788) and have included in the Discussion a new paragraph focused on the nature of our online sample and some participants’ insensitivity to the volatility manipulation (lines 477-485). As suggested, we have also verified that the participants excluded due to their insensitivity to the volatility manipulation were indeed less affected by environment when making episodic-based choices. We repeated the same analysis as in the paper of the effect of environment on these participants’ episodic-based choice index. In both the main (β=0.087, 95% *CI* = [−0.112, 0.182]) and replication (β = 0.087, 95% *CI* = [0.000, 0.168]) samples, there was a reduced and less reliable effect of environment on choice type in these excluded participants compared to the included participants.

2. Some aspects of the methods were not clear.

Thank you for these suggestions of places where the methods could be made more clear. We have made several changes to address these points, as noted below:

a. Was the order of the high and low volatility blocks counterbalanced across participants? Was the order the same in the first deck learning and second deck learning + card memory tasks? Were participants told explicitly that the environments would carry over between the two tasks? (The last point would further support using estimates fit to the first task out of the sample in the second.)

The order in which participants saw the two environments was counterbalanced across participants for the deck learning and card memory task. We have clarified this in the paper (lines 139 and 523-524). In the deck learning task, all participants first saw the low volatility environment and then saw the high volatility environment. This decision was made in order to emphasize the increased volatility of the high volatility environment relative to the low volatility environment, and this information has been added to the methods (lines 547-549). Lastly, participants were indeed told explicitly that the volatility levels of each environment would carry over from one task to the other, and we have also added this to the methods (lines 550-552). Thank you for catching that this information was not included in the methods.

b. How individual objects were re-sampled was described in the overview (lines 435-439), but I imagine the details will matter a lot to people interested in replicating and extending this work. It would help to point the interested reader to where these could be found (eg, is the code provided, or is it based on a previous study described in detail, etc.).

The sampling procedure has now been described in detail in the methods (lines 533-542).

3. Given that the results precede the methods, there are some aspects of the task that would be helpful to explain at the outset (around Figure 1). I was initially confused because the outcome in Figure 1 is $1 but the value memory was on a scale of 0-100, but this could be cleared up with a sentence about the possible outcomes. It would also be helpful to mention the mean outcome on the lucky versus unlucky deck, how frequently the lucky deck changes, and what participants are told explicitly about the volatility manipulation. This could be done in the text or with a revision to Figure 1. I was also confused about how the two samples were used until the end of the Results section (are they being analyzed together or separately? What am I looking at in Figures 2-5?). Again, a well-placed sentence at the top of the Results section would clear this up.

Thank you. We have clarified this information in multiple places throughout the text prior to the methods. In the first two paragraphs of the Results, we have added information on how often reversals occurred (lines 103-104), what participants were told (lines 105-109) and the expected value of each deck (lines 105-106). We have further added information about the types and range of outcomes to the caption of Figure 1 (lines 132-133) and information about where the main and replication sample results are reported in the introduction (lines 70-72).

4. In Figure 3, I think a slightly different comparison would be useful, in addition to or instead of the two Rescorla-Wagner models. One difference between the reduced Bayesian and RW models is that the learning rate is dynamic in the RB model but not the RW model. But another difference between the way the two models are implemented is that the RB model assumes the value of the two decks are perfectly anti-correlated (ie, it is learning only one value estimate), while the RW model does not (ie, it is learning about the two decks independently). Thus, the RB model assumes a critical aspect of the structure of the task that the RW model does not. I doubt this difference completely accounts for its better performance, but this should be tested. A δ-rule model with a fixed learning rate that learns a single value estimate (like the RB model) would be the needed comparison. This comparison would also isolate the effect of including the dynamic learning rate (according to RU and CPP) in the model.

Thank you for your idea to include this model, as we agree that it helps to isolate exactly how the RB model improves over the RW models. We added a model that implements a δ rule identical to the RW models, but with a single Q-value (labeled as RW1Q in the text). Like the RB model, this model assumes that the value of the two decks are perfectly anti-correlated, and learns over outcomes that have been re-coded in terms of the orange deck (e.g. $1 on orange is treated equivalently to $0 on blue). This model is now described in the Results of the main text (lines 228-230), listed in the Methods (line 675), and explained in detail in Appendix 3. Using a procedure identical to how the other models were fit and compared, we found that this model performed worse than both the RB and RW models we had previously presented in both samples, suggesting that the dynamic learning rate used in the RB model does indeed account for its performance improvements. The results of this comparison are reflected in an updated Figure 3A.

5. The discussion goes into the different effects of novelty, surprise, and uncertainty on subsequent memory (lines 349-364), in the context of the lack of effect of uncertainty (RU from the reduced Bayesian model) at encoding. But have the authors looked at the effect of surprise (changepoint probability in the reduced Bayesian model) at encoding? The previous studies discussed here would predict that surprise at encoding should enhance subsequent memory (and perhaps the use of episodic memory in choice). This point is not central to the manuscript, of course, but the authors have additional data relevant to the distinctions they are raising here.

This is a great suggestion, thank you. We agree that our data can provide additional insights into the effects of novelty, surprise, and uncertainty at the time of encoding on subsequent memory and episodic-memory based choice. While previously we had only investigated effects of relative uncertainty (RU) at encoding on the use of episodic memory for decisions, we additionally looked at the effects of changepoint probability (CPP) and absolute prediction error (the absolute value of prediction error; APE), which are both potential markers of surprise at the time of encoding on episodic choices. Similar to the effects of RU at encoding time reported in the Results of the main text, there was no effect of CPP (Main: β = 0.044, 95% *CI* = [−0.004, 0.092]; Replication: β = 0.004, 95% *CI* = [−0.04, 0.048]) in either sample. There was an effect of APE at encoding in the main sample (β = 0.1, 95% *CI* = [0.039, 0.165]), but this effect did not replicate (β = 0.056, 95% *CI* = [−0.013, 0.123]). Based on this, our original conclusion about the effects of these variables at encoding time on episodic-based choice remains unchanged.

In addition, based on your suggestion here along with Reviewer Three’s fifth recommendation below, we also looked at the effects of RU, CPP, and APE on participants’ performance on the subsequent memory test. Because these variables are, by definition, highly correlated with one another (e.g. at encoding RU and CPP are correlated with r=0.827), we fit multiple mixed effects regression models predicting either recognition memory (hits or misses) or value memory (response between $0-$1) for objects from each variable separately. We then performed a 20Fold leave-N-subjects out cross validation procedure to compare these models in order to determine which provided the best prediction of subsequent memory. This information is now provided in the Methods (lines 742-751 and 761-765) and the results are now mentioned in the main text (lines 342-347) and reported in Appendix 2, and in a new supplementary figure (Figure 5—Figure supplement 2). In brief, only RU at encoding time had an effect on both recognition and value memory in both samples. Specifically, higher RU at encoding predicted greater subsequent memory. Further, in both the main and replication samples, both recognition and value memory were best predicted by RU. We have now amended our discussion of the effects of surprise and uncertainty on subsequent memory to incorporate these results (lines 414-419).

Reviewer #3 (Recommendations for the authors):As mentioned in the public review, I thought this was a very interesting study and the results were clearly communicated. However, I have a number of questions/recommendations for the authors to strengthen the results and interpretation.Regarding the points made in the public review about uncertainty v volatility and context, I'd make the following recommendations:(1) Uncertainty vs. volatility (described in public review): it would make sense to reframe results around volatility rather than uncertainty, or strengthen the trial-wise RU analyses to look at trial-wise RU only at trials far away from reversals. At least there should be some discussion of where uncertainty arises besides volatility.

Thank you for this suggestion—we would like to expand on our response in the Public Review to elaborate on the variant we pursued of the specific analysis suggested here. Our understanding is that the main issue here is whether the results really are mediated by uncertainty rather than reflecting some effect of blockwise volatility other than through its dynamic effects on uncertainty. We agree with the reviewer that the trialwise RU analyses, if correctly done, could provide additional supporting evidence on this point, because the timeseries of trial-wise posterior uncertainty reflects many finer details of uncertainty, even within a block, than the cruder high-vs-low blockwise condition variable. But we think the main issue here is distinguishing blockwise effects from trialwise dynamics rather than, within trialwise effects, the dynamic effect on uncertainty of each individual ground-truth reversal event vs. other (i.i.d.) outcome variability. We are not clear what would be revealed by parsing out response to reversal events vs other noisy outcomes, since the inferential issue in this type of design (from the subjects’ perspective) is precisely that they can’t be reliably distinguished. Thus, even far from a reversal, an ideal observer will have higher posterior uncertainty in a highvolatility block due to a higher expected hazard of reversal, so even there these effects are intertwined.

Accordingly, to address one take on this, we added an additional effect of the interaction between the environment and episodic value to our combined choice model. This allowed us to look at, separately, participants’ tendency to modulate their reliance on episodic memory in response to volatility (as captured by our categorical environment variable) and in response to trialwise fluctations in posterior uncertainty (as captured by relative uncertainty), in the same model. After doing this, we found that both the environment and relative uncertainty increased sensitivity to episodic value. These changes and results are described in the Methods (lines 686-694) and Results (lines 276-277; Figure 4C).

(2) Context: I think the analyses would be strengthened with an additional model using context inference rather than incremental learning. A natural choice might be Gershman and Niv 2012, although you could possibly get away with something simpler if you assume 2 contexts.

Thank you for this suggestion. We agree that including another model to capture context inference would substantially strengthen the paper. As mentioned in our response to the related question raised in the Public Review, we have addressed this point using a hidden Markov model with two states. While the model used in Gershman and Niv, 2012 solves a similar problem, it uses a Chinese Restaurant Process to infer the total number of hidden contexts. We think it is unlikely that the participants in our task engaged in inference over the total number of contexts as they were explicitly informed that each deck could be either lucky or unlucky at a given time (essentially informing them that there were only two contexts in this task). Further, this model was developed for binary outcomes, whereas the outcomes used in our task range between $0 and $1.

(3) The focus on incongruent trials seems potentially thorny. It is intuitive why the authors do this: trials in which episodic and incremental values disagree are informative about which normative strategy the subjects are using. However, in the high volatility condition, if subjects are using incremental value, they may be more likely to have an outdated incremental value which would look consistent with the episodic choice. I would propose the authors look at congruent trials as well to confirm that they are indeed less likely to make errors on these trials in the high volatility condition than they are in the low volatility condition.

Thank you for raising this issue; we agree that it is important to disambiguate episodic-based choices from noisy choices. This point is related to Reviewer One’s first Public Review suggestion, and our solution is described in detail in our response there. In brief, we first assessed the extent to which each subject made noisier choices in the high volatility compared to the low volatility environment and then controlled for this in our analysis of episodic-based choice between environments. The effect of environment was similar to that originally reported in the manuscript following this adjustment. The reported effects (lines 178 and Appendix 1) and methods (lines 643-655) have been updated to reflect these changes.

(4) Another question relates to the interpretation of competing for episodic v. incremental strategies, as opposed to just learning about independent features. One could argue that the subjects are doing instrumental learning over objects and colors separately, and when the reliability of one feature (color) is decremented, the other feature is relatively up-weighted. This also seems consistent with the fact that episodic and incremental learning tradeoff -- attending more to the object feature would perhaps compete with color.

We agree that this is a possible interpretation of our task—for the purposes of this study, we operationalized incremental learning as a repeated feature and episodic memory as a trialunique feature, but future work can be done to more directly implicate each of these memory systems in a task that allows for them to trade off. We have added a paragraph to the paper discussing and responding to this point in more detail (lines 447-461).

(5) The authors show that uncertainty at encoding time does not have a discernible effect on the episodic index. This is evidence that volatility is modulating episodic contribution to decision-making, rather than encoding strength, which is a pretty fundamental part of the results (and in contrast to eg Sun et al. 2021, where unpredictability modulates episodic consolidation). One key thing to look at is if subjects show any difference in their ability to recall familiarity/value of objects from the different conditions. This would also speak to the question of if volatility is affecting encoding rather than just recall during decision-making. It would also make sense to look at the role of other variables at encoding time (eg prediction error) to see if these predict future use. It would also be interesting to see if subjects are storing the object value or the incremental value at the time the object was first shown -- this would be easy to check (when subjects rate the value in the last block, are they more likely to err in the direction of the incremental value at the time of encoding (eg like Figure 2A but with x-axis = incremental value at the time estimated with RW)). This would shed insight into exactly what kind of episodic strategy the subjects are deploying.

Thank you for these suggestions, as we agree that there are many other opportunities for analysis of the subsequent memory data. We now expand these analyses, as detailed below as well as in response to Reviewer Two (point five, above). First, we looked at the effects of RU, change-point probability (CPP), and absolute prediction error (APE) at encoding time on both subsequent recognition and value memory (lines 342-347 and Appendix 2).

In addition, based on your other points here, we also performed several other analyses of the subsequent memory data. We first looked at whether subsequent memory differed depending on whether an object was seen in either the low or high volatility environment. For recognition memory, this analysis consisted of calculating the signal detection metric d-prime for objects seen in each environment and testing for a difference in performance. For value memory, we tested for the presence of an interaction between an object’s true value and the environment in which it appeared on the value that was remembered by each participant. While environment did not impact value memory, recognition memory performance was better for objects seen in the high compared to the low volatility environment, suggesting that greater volatility at encoding time improved subsequent recall. These analyses are now included in the updated Methods (lines 732-751) and Results (lines 308-318) and Appendix 2. Lastly, based on your final point, we additionally looked at whether participants were more sensitive to episodic or incremental value at the time of encoding when reporting their remembered value for objects and found that object value (Main: β = 0.173, 95% *CI* = [0.159, 0.187]; Replication: β = 0.183, 95% *CI* = [0.168, 0.197]) was a substantially stronger predictor than incremental value (Main: β = 0.012, 95% *CI* = [0.001, 0.024]; Replication: β = 0.014, 95% *CI* = [0.002, 0.026]) in both samples, thereby suggesting that episodic value was more likely to drive these memory responses.

(6) The analysis showing that subjects that were better at recall in block 3 also had higher episodic index was a useful sanity check. It seems it would also be possible to perform this analysis within-subject (eg does episodic choice correlate with accurate value memory) and that would bear more on the question of whether it was uncertainty or simply a subjective preference for one strategy or another.

Thank you for this idea, which complements Reviewer One’s Public Review suggestion to sort recognition memory trials by whether the object was from episodic- or incremental-choice trials, where we found that participants have greater recognition memory for objects from episodicbased choices. We have additionally performed the within-subject analysis you suggested here by looking at whether participants better remember the value of objects from episodic-based choice trials. To do this, we fit a mixed effects linear regression predicting each participant’s subsequent memory value response from the interaction between choice type and an object’s true value (lines 752-758). We found that, in both samples, participants better remembered the value of objects from episodic-based choices. This effect is now reported in the Results (lines 308-318) and appears as a new panel in Figure 5 (Figure 5B).